

# ORCHIDEE-ROUTING: A new river routing scheme using a high resolution hydrological database

Trung NGUYEN-QUANG[1], Jan POLCHER[1], Agnès DUCHARNE[2], Thomas ARSOUZE[1],
Xudong ZHOU[1,3], Ana SCHNEIDER[2], and Lluís FITA[4]

[1]Laboratoire de Météorologie Dynamique, École Polytechnique, 91128 Palaiseau, France
[2]Sorbonne Université, CNRS, EPHE, Unité Mixte de Recherche METIS, 75005 Paris, France
[3]State Key Laboratory of Hydrology-Water Resources and Hydraulic Engineering, Hohai University, Nanjing, China
[4]Centro de Investigaciones del Mar y la Atmósfera, CONICET, UBA, CNRS UMI-IFAECI, C. A. Buenos Aires, Argentina

*Correspondence to:* Trung NGUYEN-QUANG (trung.nguyen@lmd.polytechnique.fr)

**Abstract.**

This study presents a revised river routing scheme (RRS) for the Organising Carbon and Hydrology in Dynamic Ecosystems (ORCHIDEE) land surface model. The revision is carried out to benefit from the high resolution topography provided the Hydrological data and maps based on SHuttle Elevation Derivatives at multiple Scales (HydroSHEDS), processed to a

resolution of approximately 1 kilometer. The RRS scheme of the ORCHIDEE uses a unit-to-unit routing concept which allows to preserve as much of the hydrological information of the HydroSHEDS as the user requires. The evaluation focuses on 12 rivers of contrasted size and climate which contribute freshwater to the Mediterranean Sea. First, the numerical aspect of the new RRS is investigated, to identify the practical configuration offering the best trade-off between computational cost and simulation quality for ensuing validations. Second, the performance of the revised scheme is evaluated against observations at both

monthly and daily timescales. The new RRS captures satisfactorily the seasonal variability of river discharges, although important biases come from the water budget simulated by the ORCHIDEE model. The results highlight that realistic streamflow simulations require accurate precipitation forcing data and a precise river catchment description over a wide range of scales, as permitted by the new RRS. Detailed analyses at the daily timescale show promising performances of this high resolution RRS for replicating river flow variation at various frequencies. Eventually, this RRS is well adapted for further developments in the

ORCHIDEE land surface model to assess anthropogenic impacts on river processes (e.g. damming for irrigation operation).

## 1 Introduction

Large-scale river routing is a valuable tool for validating performance of land surface models (LSMs). For example, its usefulness for quantitative verifying the revision of soil and snow hydrology in the ECMWF LSM has been shown in Pappenberger et al. (2010) and Balsamo et al. (2011). The importance of river routing schemes (RRSs) in LSMs to properly estimate the

land water storage variation was demonstrated by Ngo-Duc et al. (2007a). On the flip side, the absence of flow routing in LSM confines the evaluation of runoff simulation in medium sized catchments (Beck et al., 2016) or in an individual basin with crude estimated water residence time (Balsamo et al., 2009). This limits the benefit of river discharge measurements which provide





a unique accurate signal of the continental water cycle (Fekete et al., 2012). In addition, river routine scheme allows closing the water cycle in a fully coupled atmosphere-land-ocean model (Sevault et al., 2014; Lea et al., 2015). River discharge plays a major role in variation of surface salinity in the Bay of Bengal (Akhil et al., 2014; Jensen et al., 2016), in the Caspian Sea (Turuncoglu et al., 2013) or the formation of dense water in the northern Adriatic Sea (Vilibić et al., 2016). LSM with a proper
transfer scheme can reproduce well the decadal variability of continental water contribution to sea level change (Ngo-Duc et al., 2005b).

Simulating large-scale river flow in LSM means redistributing horizontally the surface and sub-surface runoff computed by the LSM. This horizontal redistribution in turn can feed other processes in the LSM (e.g. floodplain evaporation, irrigation). There are two popular approaches for this issue. If river process parameterization is neglected in a LSM, the first approach is
transferring runoff from the LSM to a stand-alone RRS to estimate river discharge. Of course, this transfer ignores feedback interactions between river discharge and soil hydrology of the LSM. It also accepts the loss of information as aggregating different spatial and temporal discretization between two models. One well-known runoff routing model is Total Runoff Integrating Pathways (TRIP) as well as its descendant (TRIP 2.0) (Oki and Sud, 1998; Oki et al., 1999; Ngo-Duc et al., 2007b). It has been implemented to transport runoff from the MATSIRO model (Koirala et al., 2014), the JULES 4.0 model (Walters et al., 2014)
and the HTESSEL model (Pappenberger et al., 2010; Balsamo et al., 2011). Similar solution has been applied by Decharme and Douville (2006a) to convert the simulated runoff from the ISBA model to river discharge by the MODCOU routing model for studying the impact of sub-grid hydrological parameterization on the water budget. Ducharne et al. (2003) have developed the RiTHM river routing model and applied it to 11 river basins to show the importance of parameter calibration to faithfully capture the seasonal cycle of river discharge.

The second approach is representation of river flow inside the LSM. It can be found in several state-of-the-art LSMs, such as new land model LM3 from the Geophysical Fluid Dynamics Laboratory, Community Land Model version 4.0 (CLM4) or LPJ dynamic global vegetation and hydrology model (LPJ) (Milly et al., 2014; Oleson et al., 2010; Von Bloh et al., 2010). In the LM3 model, each land cell has one river reach and water is transferred from cell to cell through a global channel network. In each river reach of the LM3 model, a non-linear relation between storage and discharge is applied. While both the CLM4
and the LPJ model use linear transport scheme which assume a global constant flow velocity. Another example is the RRS of the Organising Carbon and Hydrology in Dynamic Ecosystems (ORCHIDEE) LSM. It was designed to parameterize the river flow on continental scale (Polcher, 2003). Water transfer units (or sub-grid basins) are constructed inside each grid cell of the ORCHIDEE model, based on flow direction and watershed boundary from a global map (Oki et al., 1999; Vörösmarty et al., 2000). River basins are assembled by connecting these sub-grid basins either within or between grid cells. This routing scheme
has been applied in a large number of cases (De Rosnay et al., 2003; Ngo-Duc et al., 2005a, c; Guimberteau et al., 2012a, b).

Noticeably, a common point of these two approaches is their dependence on coarse resolution global river channel networks. Typical resolution of river map is from $1/2°$ to $1°$ (i.e. Oki and Sud (1998), Vörösmarty et al. (2000), Döll and Lehner (2002)), although Ducharne et al. (2003) relied on a $1/4°$ river map. The advent of good quality high-resolution digital elevation models (DEMs), like HydroSHEDS (Hydrological data and maps based on SHuttle Elevation Derivatives at multiple Scales, Lehner
et al., 2008), offers new opportunities to enhance river flow modelling, as pioneered by the CaMa-Flood model of Yamazaki



et al. (2011). Kauffeldt et al. (2016) notes that the evolution of LSMs demands the revision of the routing schemes to improve the adaptability to grid resolution and structure, especially since LSMs can be driven by forcing data from an atmospheric model with non-regular grid (e.g. a quasi-uniform icosahedral C-grid (Dubos et al., 2015; Satoh et al., 2014)). Wood et al. (2011) and Bierkens et al. (2015) examined the interest of hyper-resolution global LSMs, expected to function at global scale

with resolution higher than 1 km. They opined that this approach is undeniably required by societal needs and by the continuous progress of climate models. In particular, a hyper-resolution model allows providing more precise fresh water fluxes for ocean circulation simulation. It is also better suited to account for the human pressures impacting the river systems (i.e. irrigation, dam construction).

In this framework, one solution to improve the streamflow simulation aims at enhancing the quality of the river network

representation, which is made possible by the new generation of high resolution remote sensing data (Allen and Pavelsky, 2015). This study presents a preliminary attempt to revise the RRS of the ORCHIDEE LSM using a watershed description at a resolution of approximately 1 km. The new RRS is implemented and tested in the Mediterranean basin, using the data presented in Section 2. Section 3 gives a detailed description of this new RRS, and preliminary results focused on the numerical aspect are presented in Section 4. The performance of the RRS is assessed against observed river discharge in Section 5. Finally,

issues regarding the new RRS are discussed in Section 6, and short conclusions are drawn in Section 7.

## 2 Study area and simulation design

### 2.1 Study area, river discharge observations and validation metrics

The simulated domain includes 12 important rivers which flow to the Mediterranean Sea (Figure 1) and correspond to contrasted climates, from the mountainous climate leading to pluvio-nival hydrological regimes (alpine rivers) to the semi-arid climate in

Northern Africa or Southern Turkey (Ceyhan River). The Nile River was excluded as it is strongly regulated for irrigation and thus not very representative for this validation. This validation on the Mediterranean contributes directly to the HyMeX program (Drobinski et al., 2014). For each selected river, the simulated river discharge is compared to observation at the closest available station to the river mouth (Table 1). These stations have an upstream area ranging from 2000 to 95000 $km^2$. The corresponding observed discharge was gathered from 3 sources: (1) the daily and monthly river discharge time series from the Global Runoff

Data Centre (GRDC, D-56002 Koblenz, Germany); (2) the daily discharge from the French national hydrometry portal (La Banque Hydro, http://www.hydro.eaufrance.fr); (3) the daily discharge for Italian rivers (Po and Tiber) from the corresponding Italian database (personal communication by Dr. Luca Brocca, CNR-IRPI, Italy). Only seven stations provide daily discharge values. For comparison to these observations, which exhibit some data gaps (less than 15% of the observational record but for three stations, cf. Table 1), the simulated discharge values corresponding to the missing values are eliminated. This is done

to construct the studied hydrographs, as well as the derived validation metrics. Following Moriasi et al. (2007), we used the following metrics: the Pearson correlation coefficient (CC), Nash-Sutcliffe coefficient (NS), and normalized standard deviation (NSD, normalized by the observed standard deviation), which indicate good performances when reaching 1; the percent bias (PCBIAS, normalized by the observed mean discharge), the ratio of the Root Mean Square Error to the observation standard





deviation (RSR), and the cross-correlation lag time (CLT), which indicate good performances when approaching 0. The CLT is defined as the lag in in days needed to have the maximum correlation between the simulated and observed series.

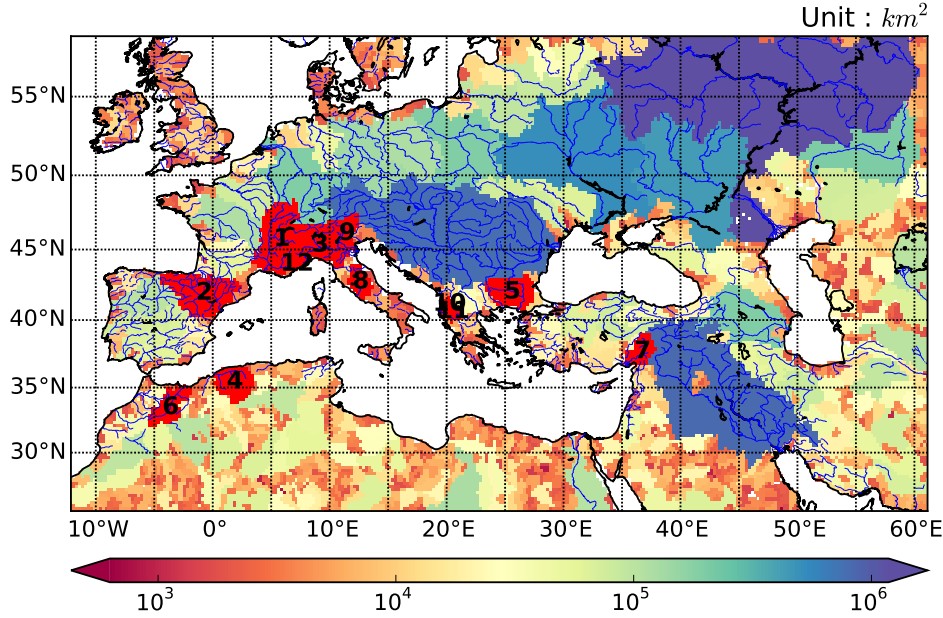

**Figure 1.** Extended simulation domain. The main watersheds are colorized as a function of maximum upstream area ($km^2$). They were extracted for an ORCHIDEE resolution of $1/4°$, with a threshold HTU number of 50. The twelve studied river basins are colored in red. They are numbered from 1 to 12, and the corresponding names are given in Table 1. The river network is plotted in blue based on the dataset from the Generic Mapping Tools (http://gmt.soest.hawaii.edu). River names are Rhône, Ebro, Po, Chelif, Maritsa, Moulouya, Ceyhan, Tiber, Adige, Shkumbinit, Devollit, Var corresponded to number from 1 to 12, respectively.

## 2.2 Numerical design

The RRS is integrated in the ORCHIDEE land surface model (Krinner et al., 2005), where the surface water budget and the

5   resulting surface and sub-surface runoff fluxes are estimated by a multi-layer soil hydrology scheme (De Rosnay et al., 2002). In this study, the model is forced by atmospheric forcing data, and three datasets are available, all based on the Watch Forcing ERA-Interim dataset (Weedon et al., 2014). They differ by the precipitation data set they are combined with to obtain bias-corrected precipitation: (1) the CRU dataset (WFDEI_CRU, Harris et al., 2014); (2) the GPCCv5 dataset (WFDEI_GPCC, Schneider et al., 2014); (3) the MSWEP dataset (WFDEI_MSWEP, Beck et al., 2017). The first two forcing data sets are at

10  $1/2°$ spatial resolution, and the last one is at $1/4°$. The simulation period extends from 1979 to 2013 after a 10-year spin-up. The time step of ORCHIDEE is 30 minutes, and the one of the RRS is 1 hour, which is compatible with the above spatial resolutions based on previous theoretical experiments (not shown).




**Table 1.** Information on stations used for validation.

| No | Station name | River | Area | Period | Data available (%) | MQ | LQ | HQ | SD |
|----|--------------|-------|------|--------|--------------------|----|----|----|----|
| 1 | Beaucaire | Rhône [a] | 95590 | 1979 - 2013 * | 100 | 1699 | 443 | 5108 | 784 |
| 2 | Tortosa | Ebro [b] | 84230 | 1979 - 1999 | 85 | 324 | 64 | 1801 | 250 |
| 3 | Pontelagoscuro | Po [c] | 70091 | 1979 - 2013 * | 99 | 1493 | 237 | 6330 | 811 |
| 4 | Sidi Belatar | Chelif [b] | 43750 | 1979 - 2001 * | 41 | 15 | 0 | 124 | 22 |
| 5 | Meric Koep | Maritsa [b] | 27251 | 1979 - 1986 * | 61 | 171 | 26 | 602 | 119 |
| 6 | Dar El Caid | Moulouya [b] | 24422 | 1979 - 1988 | 100 | 10 | 0 | 87 | 15 |
| 7 | Yakapinar (Misis) | Ceyhan [b] | 20466 | 1979 - 1986 * | 61 | 248 | 34 | 1360 | 240 |
| 8 | Roma | Tiber [c] | 16545 | 1979 - 2004 * | 86 | 185 | 73 | 626 | 91 |
| 9 | Boara Pisani | Adige [b] | 11954 | 1980 - 1984 * | 100 | 194 | 99 | 382 | 74 |
| 10 | Paper | Shkumbinit [b] | 1960 | 1979 - 1984 | 100 | 60 | 7 | 196 | 45 |
| 11 | Kokel | Devollit [b] | 1880 | 1979 - 1984 | 100 | 29 | 5 | 93 | 21 |
| 12 | Malaussene (La Mescla) | Var [b] | 1830 | 1979 - 2009 | 99 | 32 | 0 | 131 | 23 |

* Daily data is available. Area: Catchment area upstream gauging station ($km^2$).

MQ: Monthly mean discharge ($m^3.s^{-1}$). LQ: Minimum value of monthly discharge ($m^3.s^{-1}$).

HQ: Maximum value of monthly discharge ($m^3.s^{-1}$). SD: Standard deviation of monthly discharge series ($m^3.s^{-1}$).

a, b, c: Observation data source is from La Banque Hydro, the GRDC and Dr. Luca Brocca (CNR-IRPI), respectively.

## 3 River routing scheme

### 3.1 General framework

The original routing scheme in the ORCHIDEE LSM implements the linear reservoir routing method with a river network building on a 1/2° global map of the main watersheds (Oki et al., 1999; Vörösmarty et al., 2000). A brief description of this scheme can be found in Ngo-Duc et al. (2007a) and Guimberteau et al. (2012a). Each river basin is constructed by connecting a number of sub-basins which are defined inside the ORCHIDEE grid boxes, following eight outflow directions. In other words, each sub-basin represents the section of the river basin within the grid box. A sub-basin can be smaller than the ORCHIDEE grid box if the ORCHIDEE model runs at a coarser resolution than 1/2° and each grid box might encompass sub-basins of different rivers. Hereafter, we will use the term of Hydrological Transfer Unit (HTU) to designate these sub-grid basins. Runoff from one grid box can flow into neighbor one or stay in the same grid box, depending on the downstream HTUs. Thus we propose to call our scheme a unit-to-unit routing to distinguish it from the classical grid-to-grid routing.

To describe the transformation of runoff into river discharge along the river network, each particular HTU consists of three linear reservoirs with decreasing water residence times, describing the lags imposed onto groundwater flow, overland flow and streamflow (Miller et al., 1994; Hagemann and Dümenil, 1997). These lag times are the product of a slope index ($k$), characterizing the HTU, and a constant specific to the type of reservoir ($g$), calibrated by Ngo-Duc et al. (2007a). The values



of g are 0.24, 3, and 25 (day/km) for the stream, overland and groundwater reservoirs, respectively. Following Ducharne et al. (2003), the slope index is given by $k = d/\sqrt{S}$, where $d$ and $S$ are the distance and slope between a pixel and its downstream pixel. It is first defined at the $1/2°$ resolution then averaged across all $1/2°$ pixels composing a HTU. Surface runoff and drainage which are computed by the soil moisture module of the ORCHIDEE model are first lagged locally by the overland and groundwater reservoirs, then routed along the river network through a linear cascade of stream reservoirs. Applications of this scheme not only enhance the study of large-scale water balance (Ngo-Duc et al., 2005c, a) but also allow to simulate various interactions between rivers and their watersheds, such as irrigation withdrawals and flood plains (De Rosnay et al., 2003; Guimberteau et al., 2012b, a).

The availability of a high resolution DEM (digital elevation model), namely HydroSHEDS (Lehner et al., 2008), has offered the opportunity to revise this routing scheme, with the ability of constructing more adequate HTU in each ORCHIDEE grid box. This new RRS with a higher resolution river graph can operate at a fine spatial resolution (e.g. finer than 10 km) and is expected to better represent the complexity of river basins and respond better to the inhomogeneity of precipitation patterns. The next sub-section presents briefly the HydroSHEDS dataset, then the following ones elaborate on the modifications brought to the original routing scheme of ORCHIDEE to create this new version.

## 3.2 HydroSHEDS data

As a seamless near-global hydrological data set, HydroSHEDS is a suitable database for improving the river routing scheme in the ORCHIDEE model. It is available at resolutions from 3 to 30 arc-second, and provides all required information including hydrologically-conditioned elevation, drainage directions and watershed boundaries. Derived from Shuttle Radar Topography Mission, the quality of HydroSHEDS has the limitations of this radar product, returning a complex mix of terrain elevation and vegetation height, and only covering the land areas from 56°S to 60°N because of the shuttle's orbit. Despite tremendous efforts for void-filling and properly conditioning the drainage directions, some errors remain such as spurious inland sinks, and the assumption of single flow direction prevents from properly describing river bifurcations including deltas (Lehner et al., 2008). Nevertheless, this database is widely considered as the best available DEM for hydrological applications, and has shown its advantages for large-scale high-resolution river routing (Gong et al., 2011; Yamazaki et al., 2011). The stream network in the HydroSHEDS is comparable with other global hydrographic data such as HYDRO1k, ArcWorld, and the Digital Chart of the World (Lehner et al., 2006).

In this study, we use a resolution of 30 arc-second ($1/120°$ with a pixel size of 0.86 $km^2$ ca. 1 km at the Equator). It is believed sufficient for large-scale applications, and offers the possibility to complete the region north of 60°N with the HYDRO1k database at 1 km to achieve full global information (Lehner et al., 2008; Wu et al., 2012; Marthews et al., 2015). A preliminary quality control of HydroSHEDS has been performed to ensure that all pixels drain to ocean or a recognized endorheic basin. This was based on a non-exhaustive comparison with the global ESA-CCI land cover classification (Bontemps et al., 2013) to verify the existence of lakes or wetlands at inland outflow points. This procedure also provides an identifier for all the river basins with an identified outlet, in particular for the small coastal basins with a zero ID in HydroSHEDS. Based on the elevation, flow directions and basin/outlet identifier, the following information can be calculated for each HydroSHEDS



pixel for further use: slope index, flow accumulation (quantifying the amount of upstream pixels) and total downstream distance to the outlet (ocean or lake).

### 3.3 River basin and HTU construction

As mentioned above, the major improvement of the new RRS is the possibility to better describe the geometry of the river
basins by defining high resolution HTUs beneath the ORCHIDEE grid-mesh. Figure 2a underlines the major steps carried out in the river basin construction. There are two steps that control the number of HTUs in an ORCHIDEE grid-mesh which is an important issue for unit-to-unit routing with respect to the limitation of computational resources. First, in the step of constructing HTUs in each grid box, the area of each HTU is limited by an user-defined size. This allows model users to preserve as much of the hydrological information of the HydroSHEDS as they require. Second, in the last step of constructing
river network, the maximum number of HTUs in an ORCHIDEE grid box is setting by an user-defined number, which results in changing the average HTUs area over the ORCHIDEE grid-mesh. The construction of the HTUs is illustrated in Figure 2b for the Rhône River (98,000 $km^2$ in Switzerland and South-Eastern France, with an outlet in the Mediterranean Sea near Marseille) and an ORCHIDEE grid-mesh resolution at the 1/8°. For clarity, we impose here a maximum number of nine HTUs per ORCHIDEE grid box, but the optimal HTU number is discussed in section 4. In this framework, the number of upstream
HTUs contributing water to each ORCHIDEE grid cell is depicted in Figure 2b. Logically, the ORCHIDEE grid cells which cover the main stream of the Rhône River have large number of upstream HTUs, so this stream can be recognized in dark blue while smaller tributaries are identified in yellow. This river basin design depends on the arrangement of HTUs inside each ORCHIDEE grid box, as exposed by a simple example in Figure 2c-d. These two figures explain the definition of HTUs in the ORCHIDEE grid cell which is marked by the orange rectangle in the Figure 2b.

The first step consists in identifying all the outlets from the grid box, and all HydroSHEDS pixels sharing the same grid box outlet are combined to build preliminary HTUs. For example, the borders of these preliminary HTUs is delineated by orange lines in Figure 2c. After this step, a preliminary HTU can be larger than the user-defined size (e.g. an user can set the maximum area of a HTU is smaller than 2% of the ORCHIDEE grid box). For instance, the preliminary HTU with the outlet point marked by orange circle in Figure 2c will cover about 80 % area of the grid box. Hence, a procedure was developed to partition these
large preliminary HTUs to the user-defined size. The partitioning process relies on the Pfafstetter topological coding system for streams and basins (Pfafstetter, 1989; Verdin and Verdin, 1999): the flow accumulation is used to identify the main stream of the HTU to partition, and its main four tributaries; this results in dividing the large HTU into nine smaller HTUs comprising the basins of the four tributaries and five inter-basins. The hexagons in Figure 2d denote the outlet points of the newly divided HTUs (called inter-HTUs), which are identified by distinct colors in this figure. All these coloured HTUs initially belong to
the preliminary HTU flowing out of the grid box at the outlet point marked by the orange circle on the eastern edge. The division process will continue if the HTU size is larger than the user-defined size, which can obviously not be smaller than the HydroSHEDS pixels (ca. 0.86 $km^2$).

While the delineation based on the Pfafstetter codification informs on the connectivity between the HTUs deriving from the same preliminary HTU (thus inside one ORCHIDEE grid box), the linkage between HTUs belonging to different grid





boxes needs a supplementary procedure. After a HTU is constructed, it gets a unique identifier, and the corresponding HTU outlet is located (as the pixel with the biggest flow accumulation). We further calculate the total area upstream from the HTU outlet, and the distance from the HTU outlet to the river basin outlet (ocean or endorheic lake). A complex procedure uses this information to identify the downstream HTU in neighbor grid boxes, then re-establish the connectivity and coherence of the

river network. The advantage of this procedure is the possibility to function on both regular and non-regular grids. Finally, the last step not only ensures the final number of HTUs does not exceed the user-defined threshold (e.g. 9 HTUs per grid box in the case of Figure 2b) but also balances the HTU areas to avoid too strong asymmetry of HTUs area distribution. This is done by combining the smallest HTUs which flow out of the ORCHIDEE grid box. Note that the number of tiny HTUs increases remarkably as the ORCHIDEE resolution decreases (e.g. coarser than 1/4 °). For example, there are tiny preliminary HTUs

which only include one HydroSHEDS pixel, as marked by orange squares in Figure 2d.

The river basin construction described above is the solution for the main limitation of the old RRS which is its poor representation of small river catchment, i.e. with area under 2500 $km^2$ (1/2° × 1/2°). The reason for this shortfall is that the old RRS implements the river network with a basin description at 1/2° resolution (Oki et al., 1999; Vörösmarty et al., 2000). Figure 3a compares the simulated areas of 12 river basins in this study to a reference area, which is from the global river network data at

1/16° spatial resolution (Wu et al., 2012). The blue triangles and green circles denote the modeled area in the new RRS with the ORCHIDEE resolution of 1/2° and 1/4°, respectively. The modeled areas in the old RRS with the resolution of 1/2°are shown by the orange diamonds. The numbers correspond to the river names are given in Table 1. Significantly, the new RRS accurately constructs the river basin area of all 12 rivers (i.e. the CC values are 0.99 and 1.0). The old RRS only represents well the area of the Rhône (1), the Ebro (2) and the Maritsa River (5). While the errors are high for other rivers such as the Po (3),

the Chelif (4), the Ceyhan River (7). In addition, the old RRS can not represent four small river basins (i.e. the Adige (9), the Shkumbinit (10), the Devollit (11) and the Var River (12)). Figure 3b shows the description of the Tiber River basin (in Italy, with an outlet to the Tyrrhenian Sea at Rome) with the old RRS and a regular latitude-longitude grid at a resolution of 1/2°. This river basin covers only 4 grid boxes with maximum upstream area about 9100 $km^2$, whereas the upstream area of station Rome is about 16545 $km^2$ according to the GRDC database. In this case, the old RRS misses the southern part of the Tiber

river basin. The simulated outlet point (shown by orange circle) of the Tiber River is located about 70 km from the coastline. At the same resolution of 1/2° the new RRS depicts more satisfactorily the southern part of the Tiber River (Figure 3c). The total area is about 13600 $km^2$ with the outlet at about 30 km from the coast. As the ORCHIDEE resolution increases, the representation of small river becomes more realistic with the new RRS. An example is shown on a regular latitude-longitude grid at 1/16° in Figure 3d. The total area of the Tiber river reaches nearly 16600 $km^2$ and the river mouth is placed properly

at the coast. The position of the mouth can vary from one resolution to the other because of changing land-sea mask in the ORCHIDEE model.

The aim of this study is to develop a new RRS for the ORCHIDEE LSM which can simulate a wider range of river catchment sizes. In this study, we focus on rivers which contribute significant amounts of freshwater to the Mediterranean Sea. Because of the complex topography and the diversity of climates surrounding the Mediterranean, a wide range of catchment sizes need

to be covered (from about 2000 to 95000 $km^2$, cf. Table 1) in order to represent a realistic water input into this internal sea.





Due to the limitation of the old RRS with small catchments (Figure 3), comparison of simulation quality between the old and new RRS is difficult in this region. Therefore, the following sections focus on understanding the new RRS only.

### 3.4 Water routing process

Surface and sub-surface runoff from an ORCHIDEE grid box are distributed to the overland and groundwater reservoir of the
embedded HTUs proportionally with their areas. As in the original routing scheme, runoff is routed downstream with a delay time that is controlled by the number of HTUs along the stream, and the properties of each HTU, namely their slope index $k$ and reservoir parameter $g$, the product of which defines the time lag of each HTU. The slope index is first calculated at the 1-km resolution based on the slope and length of the HydroSHEDS pixels, and it needs to be properly aggregated at the HTU scale. When used with the new topographic information based on HydroSHEDS, the simple averaging performed in the
original version of the routing scheme leads to consider a water travelling distance of 1 km, which is underestimated for most HTU sizes. We developed a new algorithm that uses the drainage directions and the resulting distance of each pixel to the HTU outlet. In practice, for each pixel, we define $K$ as the sum of all the 1-km values of $k$ along the corresponding downstream line. The upscaled value of $k$ for the HTU is then given by the average of $K$ across all the pixels composing the HTU. It can be shown this corresponds to the combination of the different pixels with a unit hydrograph modelling. As a result, the slope index
of HTUs changes with the area and length $D$ of the HTUs, so that the streamflow velocity (given by $Kg/D = g/(sqrt(S))$) does not depend, or weakly, on the HTU scale. In addition, the three reservoir parameters ($g$) have been recalibrated, leading to values of 0.01, 0.5 and 7.0 (day/km) for the stream, overland, and groundwater reservoirs respectively, i.e. smaller than with the former $1/2°$ topographic data. These values were estimated empirically for the Rhône river basin and implemented over the entire simulated domain.

**4   Impact of HTU size**

### 4.1 Experiment design

As already mentioned, the number of HTUs in each ORCHIDEE grid box increases the computing requirements, but it is also expected to increase the simulation quality, owing to a better description of the river flow directions and basin boundaries. Therefore, we investigate here the impact of the average size of HTUs on the simulated hydrographs, in order to better un-
derstand the numerical aspects of the new RRS, and find the best compromise between computational needs and simulation quality. To this end, the ORCHIDEE model is first used without the RRS to generate surface and sub-surface runoff at the $1/4°$ spatial resolution, using the WFDEI_MSWEP atmospheric forcing from 1979 to 2013 (section 2). These fields are then interpolated to four other horizontal resolutions of approximately $1/16°$, $1/12°$, $1/8°$ and $1/2°$, using the first order conservative remapping module of the Climate Data Operators (CDO, Schulzweida, 2018). Then, these five sets of runoff data are used to
simulate river discharge with the new RRS. For each resolution, a reference case is chosen to preserve most of HydroSHEDS information. This would ideally be achieved with an average HTU size of 0.86 $km^2$, but we used higher values at the coarsest





resolutions (1/4°and 1/2°) to limit the computing requirements. As a result, the average HTU areas of the reference cases are approximately 0.86, 0.86, 0.97, 2.60 and 8.20 $km^2$ for resolutions of 1/16°, 1/12°, 1/8°, 1/4°and 1/2°respectively. These reference HTU areas vary between 2.0 and 0.3 % of the grid box areas. Finally, for each resolution, several alternative RRS simulations are performed by varying the average HTU size from 0.3 % to about 80 % of the ORCHIDEE grid box areas.

## 5  4.2  Results

The analysis of these experiments focuses on 3 stations: Beaucaire (on the Rhône River, with an upstream area of 95590 $km^2$), Pontelagoscuro (on the Po River, 70091 $km^2$) and Roma (on the Tiber River, 16545 $km^2$).

In Figure 4, each line corresponds to a different resolution and shows the Nash-Sutcliffe coefficients between the reference case and simulations with larger HTUs. It is interesting to note that, for each resolution, the simulation results degrade when
the ratio of HTUs average size to area of ORCHIDEE grid box (further abbreviated as area-ratio) exceeds a certain value. The 1/2°simulations start to deviate remarkably from the reference case when the area-ratio exceeds 0.7%. For resolutions of 1/4, 1/8, 1/12 and 1/16 °, the downgrade points are at area-ratios of 1.5, 2.0, 3.0 and 3.0 %, respectively. Another remarkable point is that the degradation is weaker at the monthly than at the daily timescale. At the monthly timescale, the Nash-Sutcliffe coefficient with respect to the reference simulation remains above 0.7, while it drops to negative value or near zero at the daily
timescale. This highlights that the simulation results are more sensitive to the resolution and variation of HTUs arrangement at the daily timescale. Nevertheless, the degradation occurs at the same area-ratio as for both timescales.

These thresholds of area-ratio can be explained by analyzing the distribution of HTUs areas for each resolution. Figures 5a,c,e reveal the skewness of this distribution for the Rhône, Po and Tiber rivers. For all three rivers, the skewness peaks at these thresholds of area-ratio. Skewness indicates the lack of symmetry produced by the existence of a few much larger HTUs
among plenty of smaller ones. The large HTUs appear due to the combination procedure involved to control the number of HTUs. As we move to larger area-ratios, the large and more equal HTUs start to dominate, reducing the skewness of the distribution, and degrading the simulated flow by lack of details in river graphs and basin characteristics at small area-ratios.

Figure 5b,d,f shows the decrease of total number of HTUs as average size ratio of HTUs increases. The blue lines display steep decreases of the number of HTUs in the case of 1/16° resolution. For the Rhône River, it decreases from around 120,000
to 3,000 HTUs. For a small river such as the Tiber, it also drops from about 18,000 to 500. In contrast, this range for the case of 1/2° resolution is only about 11,000 for the Rhône river and 2,000 for the Tiber. The change of total catchment area and the HTUs slope factor are also investigated, but their impacts on the behaviour of the new RRS is small (not shown). The simulated basin area only varies strongly in the case of the 1/2° resolution grid. The catchment construction with resolution higher than 1/4 ° gives more steady river areas. The examination with other metrics (i.e. correlation coefficient, root mean square error,
standard deviation and kurtosis of the HTU size distribution) supports the following analysis with similar signals thus it is not shown here.

Figure 6 highlights another aspect of the impact of HTUs size on the simulated river discharges, by focusing on the performances of the simulations against observed discharge. With an average size of HTUs maintained at approximately 13 $km^2$ (corresponding to an area-ratio ranging from 0.5% at 1/2° to 30% at 1/16°), this figure compares monthly simulated river




discharge series with the observation data as the ORCHIDEE resolution increases. Therefore, this figure provides a simple investigation on the dependency of the new RRS on the ORCHIDEE resolution. The comparison is focused on two metrics, the Pearson correlation coefficient (CC) and Nash-Sutcliffe coefficient (NS), calculated at the monthly timescale. They indicate satisfactory performances, with a good stability at most resolutions. The only decrease in performance is found at Beaucaire

at $1/2°$. Because the runoff fluxes were interpolated from the 1/4 °to other resolutions, this test does not include the impact of the resolution change on the other components of ORCHIDEE (water and energy budgets). If the full ORCHIDEE was run at these resolutions, we expect the changes in runoff and drainage to be larger than the impact of resolution on the routing demonstrated here. The main point is the stable performance of the RRS at a fixed HTUs size (i.e. 13 $km^2$) as the ORCHIDEE resolution varies. Although the new RRS inherently depends on the ORCHIDEE resolution as implementing a unit-to-unit

routing method, it can be seen that with a certain average size of HTUs, a stable simulated quality is expected for a wide range of ORCHIDEE resolutions.

### 4.3   Practical HTU size

At a given resolution of the ORCHIDEE model, a higher area-ratio (ratio of HTUs average size to ORCHIDEE grid box area) means less HTUs thus less computational requirements. On the other hand, the above analysis on the numerical behaviour of

the new RRS indicates that the quality of the simulations deteriorates at large area-ratios, and that this degradation starts at the area-ratio corresponding to the highest skewness of the HTU size distribution. It can thus be assumed that the best compromise between simulation quality and computational cost is achieved at this practical HTUs size, which seems rather constant across the studied basins. Here, for the 1/2 ° resolution, we find a practical area ratio of 0.7%, corresponding to a practical average size of HTUs of about 22 km$^2$; for the 1/4 ° resolution, the practical area ratio is 1.5%, corresponding to a practical average

size of HTUs of about 11 km$^2$. This allows a strong computational gain compared to RRS simulations that would be done at the highest possible resolution (ca. 0.86 km$^2$), and a gain by a factor of 10 is obtained for the recommended resolution.

### 5   Routing scheme performance

### 5.1   Experiment design

The goal of this section is to evaluate the performance of full ORCHIDEE simulations against river discharge observations.

To this end, the new RRS is fully coupled to the ORCHIDEE LSM, which is driven by the three atmospheric forcing datasets presented in section 2: the WFDEI_CRU and WFDEI_GPCC with a spatial resolution of $1/2°$ and the WFDEI_MSWEP with a spatial resolution of 1/4°. The three corresponding simulations are performed with practical average HTU sizes identified above, viz. 22 $km^2$ at $1/2°$ and 11 $km^2$ at 1/4°. For comparison, similar simulations are also performed with different average HTU sizes (from 0.3% to 50% of the grid box area), and also with the old RRS. In this case, the ORCHIDEE simulations are only

run with the two $1/2°$ forcing datasets, owing to the limitations of the OLD RRS at finer resolutions. All the simulations extend from 1979 to 2013 after a 10-year spin-up.



## 5.2 Monthly timescale

Figure 7 shows the annual cycle of simulated and observed discharge for stations Beaucaire (the Rhône river) and Ponte-lagoscuro (the Po river). The new RRS captures satisfactorily the seasonal cycle of observed discharge in the two stations. For Pontelagoscuro, the simulations reproduce adequately two characteristic peaks of the pluvio-nival regime, in October and May.

For Beaucaire, the new RRS not only captures the high peak in January but also the gradual decrease to low flow in August, except with the WFDEI_CRU data. Most simulations also display a positive bias, which can be attributed to excessive runoff fluxes (since the RRS is conservative and does not change the long-term mean discharge), and explained by systematic errors in the water budget parametrization of ORCHIDEE or biases in the three forcing data sets. The purple boxplots present the sensitivity of simulated discharge to the average HTU size (from about 0.3% to about 50% of the grid box area) and quantify

the uncertainty coming from the numerical choices of the scheme. As the HTUs average size varies, the simulated discharge fluctuates in a range which is much smaller (about 40 % smaller) than the magnitude of the bias when compared to observation. In other words, it can be said that the numerical uncertainty is small compared to the uncertainty in the forcing data. According to Beck et al. (2017), the quality of precipitation data from MSWEP is overall better than WFDEI_CRU when compared with observations from 125 FLUXNET stations. This is confirmed by the lower magnitude of the bias in the case which uses

WFDEI_MSWEP (Figure 7 e,f), while this error, when using WFDEI_CRU, is larger at both stations (Figure 7 a,b). We can thus confirm that accurate atmospheric forcing is an important factor which determines the performance of a RRS, as already reported by many studies (e.g. Ngo-Duc et al., 2005c; Guimberteau et al., 2012a; Pappenberger et al., 2010). In addition, the simulation results with the old RRS (shown with the orange lines) confirm that the new RRS better matches the observed discharge, which is likely related to its better estimation of the river basin area and structure.

Validation at 12 stations highlights that the new RRS simulates acceptable river discharge at monthly timescale (Figure 8 and Table 2). The Taylor diagrams in Figure 8 present the CC and the NSD between simulation and observation. The value of CC and NSD are both 1.0 at the black star point, where simulated results would have same amount of variation and perfect linear correlation with observation. For this reason, simulations that agree well with observation will lie close to this reference point. Over the 12 stations, the CC are all above 0.6 and the NSD are in a range of 0.5 to 1.5. The new RRS achieves the best

performances at stations Beaucaire (1), Pontelagoscuro (3), Yakapinar Misis (7), Kokel (11), Paper (10), Malaussene (12), and Tortosa (2). It is interesting that these stations display a wide range of upstream areas, with monthly mean discharge from about 30 to 1500 $m^3 s^{-1}$ (Table 1). The worse overall results are found at stations Boara Pisani (9), Dar El Caid (6), Roma (8), Sidi Belatar (4), and Meric Koep (5). For Boara Pisani (Adige River, Northern Italy), NSD values of the experiment WFDEI_CRU and WFDEI_MSWEP are both higher than 2.2, which is linked to the high positive bias with these forcing datasets (i.e.

the estimated discharge is about twice higher than observation, as shown in Table 2). As the overestimation in experiment WFDEI_GPCC is smaller (i.e. the PCBIAS is about 22%), NSD value stays around 1.10. It should be noted, however, that CC values for this station range from 0.77 to 0.94, so the monthly variability is quite well captured. In particular, the simulated monthly series reproduce well the observed hydrological regime with rather weak seasonal variations and two moderate peaks (in June and October). But most of the poor performances are found in the driest part of the Mediterranean basin. The negative



bias at Roma can be attributed to the underestimated runoff during summer time (July to September) or water management practices. The underestimation of discharge is even worse at Sidi Belatar, which belongs to the longest river in Algeria and rises from the Saharan Atlas. The observed annual mean discharge is lower than 20 $m^3 s^{-1}$, with very low values in summer time (close to zero), and none of the simulations can reproduce this characteristic, whichever the forcing dataset. The time

series of the anomaly of monthly discharge (with respect to the mean seasonal cycle) are also analyzed in Figure 8b to assess the inter-annual variability of discharge. It shows about the same perspective as the monthly series with lower CC values, although good CC (about 0.9) are still found at Beaucaire (1), Pontelagoscuro (3), Yakapinar Misis (7). The errors at station Boara Pisani (9), Sidi Belatar (4) and Dar El Caid (6) are more clearly demonstrated. Regarding the effect of different forcing data, no clear hierarchy is found, although the above findings confirm that the biases of surface and sub-surface runoff by the

ORCHIDEE LSM, at least partly due to the biases of the forcing datasets, have a strong impact on the simulated streamflow. This is probably the main reason for the low values of NS and high absolute values of PCBIAS (shown in Table 2).

It is important to note that the impact of human regulation on natural streamflow is neglected in the current version of the new RRS. As a result, irrigation is not represented at all in these simulations while it is known to play an important role in the Mediterranean region (Margat and Treyer, 2004). According to Montanari (2012), the annual water withdrawal for irrigation in

the Po River basin is about 17 km³, i.e. a third of the mean annual discharge (47 km³). Not all withdrawals are transformed into evaporation and thus a part of these abstractions will return to the river. Nevertheless, one can assume that the observations for Pontelagoscuro displayed in Figure 7 probably underestimate the natural river discharge. Snoussi et al. (2002) also underline that the water discharge at the Moulouya River (station Dar El Caid) has been reduced by almost 50% due to the construction of Mohamed-V reservoir in 1967, which could explain the large positive biases of our simulations. Figure 9a shows that, in the

Ebro basin (Spain), the discharge peaks in May and June are largely overestimated by the simulations, whichever the forcing dataset. This is not the case, however, when referring to the observed record from 1920 to 1930 (orange line). At this period, human impacts on the natural Ebro river flow were not significant and the discharge peak from May to June was stronger than the one associated to the winter rains. This analysis strongly supports that the overestimation of river discharge over the last decades could be alleviated by a proper representation of human water abstractions for irrigation in the new RRS.

For the Maritsa river, Artinyan et al. (2007) noted that about 7% of the riverflow is used for irrigation during summertime (June to August). Despite the fact that this study only covers the year 1996, their findings nevertheless underline the role of anthropogenic influence on river processes, which probably contribute to the positive bias of our simulations. Yet, in contrast to the Ebro, the spread in river discharge bias caused by the forcing data uncertainty is larger than the bias itself, suggesting that the role of human processes is not as large as for the Spanish basin. The design of the new RRS in the ORCHIDEE

LSM, because of the high level of hydrological information at sub-grid level, is a good starting point for including these anthropogenic pressures, which are usually distributed at a small-scale. It lays the foundation to characterize their impact on the simulated discharge based on high-resolution maps, as available for irrigation (Siebert et al., 2005; Portmann et al., 2010) or dams (Lehner et al., 2011; Beaufort et al., 2017) .





### 5.3 Daily timescale

Simulation quality at daily timescale is validated at 7 stations, listed in Table 1. Table 3 presents five metrics considering the variability of daily time series (i.e. CC, NS, NSD, CLT) and the magnitude of the error (i.e. RSR). The daily CC values are slightly smaller than the monthly CC values, but remain higher than 0.6 for 5 stations, where the short-term variabil-

ity of streamflow is correctly reproduced. The best result can be found for station Pontelagoscuro with forcing data from WFDEI_GPCC (Table 3). The two exceptions are Roma (Tiber) and Sidi Belatar (Chelif), which were also displaying weak CC at the monthly timescale. Table 3 also gives values of CLT in a range from -25 to 1 days, which suggests that some CC values can be improved by accelerating the transfer water in the RSS. We also find that the daily NS are much weaker than the monthly values. At the daily timescale, acceptable NS, which are usually taken as higher than 0.5 (e.g. Moriasi et al., 2007;

Tavakoly et al., 2017), are only found at stations Pontelagoscuro (Po) and Yakapinar Misis (Ceyhan), which are the stations maximizing the monthly NS. The reason is that NS is very sensitive to mass balance errors, i.e. river discharge bias (Krause et al., 2005; Gupta and Kling, 2011), and to daily time series of small dry rivers (Schaefli and Gupta, 2007). Although NS is very commonly used to evaluate hydrological simulations, this criteria still raises many questions about its application (Criss and Winston, 2008; Gupta et al., 2009). The larges biases also explain why all RSR values in Table 3 are larger than 0.5.

The large fluctuations of all the metrics between the simulations with 3 different forcing again highlight the importance of the accuracy of the atmospheric conditions imposed on the land surface model.

For the four stations with the best daily NS, Figure 10 visualizes the simulation results at the daily time scale using flow duration curves, which represent the percentage of time over which discharge exceeds the discharge value indicated on the vertical axis. This analysis shows that discharge at Beaucaire (Fig. 10a) is overestimated over the full range of frequencies,

except with WFDEI_MSWEP, which induces lower flows than observed for the 20% lowest flows (exceedance frequency above 80%). Except with WFDEI_CRU, realistic statistics are found at almost all frequencies at Pontelagoscuro (Fig. 10b), explaining the good results presented in Table 3. At Yakapinar Misis (Fig. 10c), in contrast, the RRS captures well the low flows under 500 $m^3.s^{-1}$, but the the highest flows which happen less than 10% of time are strongly underestimated. The poor performances at station Sidi Belatar are confirmed by Fig. 10d. At this station, stream flow exceeds 100 $m^3.s^{-1}$ occur only 5%

of time and more than 50% of times, the flow is close to zero. It is difficult to capture the daily discharge at this station, and the large agricultural water demand over the Chelif basin (Mahe et al., 2013) probably contributes to the errors. But the main error source seems to be the forcing datasets, since the observations fall in their wide uncertainty range.

Stream flow fluctuations possesses alternative frequencies in addition to the daily and monthly oscillation. This can be seen in the power spectrum of daily river discharge at Beaucaire in Figure 11a. The power sprectrum corresponds to the squared

amplitude at each frequency and was extracted using discrete Fourier Transform and smoothed with a Savitzky-Golay filter. Trend lines for high frequency variation below 30 days and low frequency above 30 days are also plotted. Based on the previous diagnostics we expect the new RRS to replicate well the variation of river flow at various frequencies. Figure 11a shows the good match of the two power spectrum patterns. The $\beta$ values for the simulation are 0.91 and 2.39 at frequencies above and below 30 days, respectively, very close to the corresponding slopes for the observations, i.e. 0.81 and 2.28.





At high frequency, there is a mismatch of power density around the 3 day period. It can be traced back to the simulated sub-surface runoff, which shows a similar peak at the same period (Figure 11b). The peak at 3 days is probably characteristic of the soil moisture diffusion scheme of the ORCHIDEE model, as it does not have a signature in precipitation and evaporation averaged over the entire Rhône catchment Figure 11). The spectrum of these two fluxes which characterize the exchange with

the atmosphere are more noisy at the synoptic scales (below 15 days) and display a stronger slope for low periods. This results is indicative of a too fast soil moisture diffusion in ORCHIDEE, or an insufficient buffering of the resulting sub-surface flow (drainage) by the routing reservoir representing groundwater flow. It further suggests that the link between the soil hydrology and the routing scheme is too simple in ORCHIDEE and lacks an appropriate representation of the aquifers. Human activities and the regulation of river flows also affect the river discharge variability at many frequencies (e.g. hydropeaking as a result of

the storage for hydropower plants (Meile et al., 2011)) and thus the validation of this variability either requires the analysis of pristine catchments or the representation of water management infrastructures in the model.

**Table 2.** Evaluation metrics for monthly river discharge simulations by the ORCHIDEE model with the new RRS. NS: Nash–Sutcliffe efficiency, PCBIAS: percent bias and RSR: ratio of Root Mean Square Error with observation standard deviation.

| No | Station name (river) | NS [−] | | | PCBIAS [%] | | | RSR [−] | | |
|---|---|---|---|---|---|---|---|---|---|---|
| | | (1) | (2) | (3) | (1) | (2) | (3) | (1) | (2) | (3) |
| 1 | Beaucaire (Rhone) | 0.18 | 0.52 | 0.54 | 23.92 | 18.92 | 12.87 | 0.91 | 0.69 | 0.68 |
| 3 | Pontelagoscuro (Po) | 0.51 | 0.83 | 0.70 | 21.06 | -9.01 | -5.96 | 0.70 | 0.42 | 0.55 |
| 4 | Sidi Belatar (Chelif) | 0.02 | 0.06 | -12.34 | -63.68 | -68.18 | 338.22 | 0.99 | 0.97 | 3.65 |
| 5 | Meric Koep (Maritsa) | -0.11 | 0.31 | -2.81 | 47.32 | 32.07 | 112.82 | 1.06 | 0.83 | 1.95 |
| 7 | Yakapinar Misis (Ceyhan) | 0.52 | 0.58 | 0.80 | -2.56 | 1.44 | -10.60 | 0.69 | 0.65 | 0.45 |
| 8 | Roma (Tiber) | -0.08 | -0.18 | -0.38 | -7.62 | -7.42 | -21.76 | 1.04 | 1.08 | 1.17 |
| 9 | Boara Pisani (Adige) | -11.56 | -0.09 | -7.70 | 116.87 | 22.28 | 85.87 | 3.54 | 1.04 | 2.95 |
| 2 | Tortosa (Ebro) | -0.26 | 0.02 | -0.03 | 54.05 | 59.68 | 34.69 | 1.12 | 0.99 | 1.02 |
| 6 | Dar El Caid (Moulouya) | -2.15 | -2.30 | -3.05 | 201.98 | 204.16 | 16.72 | 1.77 | 1.82 | 2.01 |
| 10 | Paper (Shkumninit) | 0.20 | 0.39 | 0.26 | -45.90 | -40.28 | -47.60 | 0.89 | 0.78 | 0.86 |
| 11 | Kokel (Devollit) | 0.59 | 0.72 | 0.17 | -17.27 | -14.17 | 14.64 | 0.64 | 0.53 | 0.91 |
| 12 | Malaussene (La Mescla) (Var) | 0.18 | 0.25 | 0.24 | -20.85 | -38.75 | -27.68 | 0.91 | 0.87 | 0.87 |

(1), (2), (3) are simulation results with forcing data from WFDEI_CRU, WFDEI_GPCC, WFDEI_MSWEP, respectively.

## 6   Discussion

The first attempt to model hydrology on the global scale comes from the atmospheric science community (Manabe, 1969). Nowadays, global hydrology has been considered as an essential component of LSM and Earth system model (Bierkens,

2015). To close the global water cycle, representation of the lateral transport of water is recognized as an important topic in





**Table 3.** Evaluation metrics for daily river discharge simulations by the ORCHIDEE model with the new RRS. CC: Pearson correlation coefficient, NS: Nash–Sutcliffe efficiency, NSD: Normalized standard deviation and CLT: Cross-correlation lag time, RSR: ratio of Root Mean Square Error with observation standard deviation.

| Station name (river) | CC [−] | | | NS [−] | | | NSD [−] | | | CLT [day] | | | RSR [−] | | |
|---|---|---|---|---|---|---|---|---|---|---|---|---|---|---|---|
| | (1) | (2) | (3) | (1) | (2) | (3) | (1) | (2) | (3) | (1) | (2) | (3) | (1) | (2) | (3) |
| Beaucaire (Rhone) | 0.59 | 0.69 | 0.67 | 0.01 | 0.26 | 0.08 | 1.00 | 1.01 | 1.24 | -6 | -10 | -5 | 0.99 | 0.86 | 0.96 |
| Pontelagoscuro (Po) | 0.79 | 0.86 | 0.68 | 0.40 | 0.71 | 0.38 | 1.15 | 0.98 | 0.95 | -1 | -1 | -5 | 0.77 | 0.54 | 0.79 |
| Sidi Belatar (Chelif) | 0.29 | 0.37 | 0.38 | -0.05 | 0.05 | -8.50 | 0.56 | 0.51 | 3.00 | 1 | 1 | -1 | 1.03 | 0.98 | 3.08 |
| Meric Koep (Maritsa) | 0.61 | 0.54 | 0.60 | -0.51 | -0.12 | -2.40 | 1.37 | 1.07 | 1.61 | 0 | -13 | -9 | 1.23 | 1.06 | 1.84 |
| Yakapinar Misis (Ceyhan) | 0.60 | 0.64 | 0.79 | 0.36 | 0.41 | 0.61 | 0.56 | 0.60 | 0.84 | -25 | -21 | -2 | 0.80 | 0.77 | 0.62 |
| Roma (Tiber) | 0.39 | 0.44 | 0.51 | -0.21 | -0.22 | -0.47 | 0.97 | 1.08 | 1.30 | -12 | -11 | -2 | 1.10 | 1.10 | 1.21 |
| Boara Pisani (Adige) | 0.70 | 0.66 | 0.62 | -6.83 | 0.04 | -5.99 | 2.14 | 1.12 | 2.50 | -7 | -7 | 0 | 2.80 | 0.98 | 2.64 |

(1), (2), (3) are simulation results with forcing data from WFDEI_CRU, WFDEI_GPCC, WFDEI_MSWEP, respectively.

LSM development as it contributes directly to closing the water cycle in Earth system models (e.g. Arora et al., 1999; Ngo-Duc et al., 2007b; Gong et al., 2011; David et al., 2011). Moreover, recent developments in river routing modelling have allowed to investigate the impact of hydrological processes (e.g. floodplains d'Orgeval et al. (2008)) and groundwater interactions (Vergnes et al., 2014) on the climate system, and conversely, the impacts of human regulation and climate change on natural

streamflow (Voisin et al., 2013b, a). A strong emphasis has also been given on high resolution routing schemes (Ducharne et al., 2003; Yamazaki et al., 2011; Zhao et al., 2017). In this line, the aim of the present study is to make the original RRS in the ORCHIDEE LSM compatible with the HydroSHEDS data. We believe that we have designed an innovative infrastructure which will be the basis to further studies regarding the links between the global water cycle and anthropogenic impacts (human regulation as well as climate change) or hydrological processes.

The key advantage of this new scheme is its ability to provide a precise river catchment description over a wide range of scales. Figure 1 shows for the Mediterranean area that not only the Danube, with a catchment of about 800,000 km$^2$, but also the nearly 2,000 km$^2$ Var river basin are represented. Andersson et al. (2015) denoted that accurate river basin area is the first factor for properly simulating river discharges. Arora et al. (2001) also remarked the difficulty of reliably simulating river flow for small basins at large spatial scale. The proposed scheme provides a good river network quality over a variety of resolutions

and simulation scales, thus reducing the need for network-response function to reduce this scale dependency Gong et al. (2009). The new RRS preserves as far as possible the hydrographic details of the 1 km HydroSHEDS data inside each grid box of the ORCHIDEE LSM. It not only represents more accurately the total catchment area but also locates more precisely the river

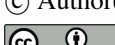


mouth at the coast. This will be an asset as coupling the ORCHIDEE LSM into a global or regional Earth system model. In addition, the new RRS can operate on unstructured grids with resolution up to the order of 1 $km^2$. This flexibility is clearly an advantage of this RRS compared with other large-scale hydrological models (Kauffeldt et al., 2016). In addition, Verdin and Verdin (1999) highlighted the value of the Pfafstetter codification to preserve river network topology and topographic control

of drainage. It supplies a spatial framework which reconciles information from scale of general circulation models GCMs to smaller scale river processes (e.g. irrigation operation). Therefore, it provides adaptability to higher resolution of input data (e.g. the 3 arc-second HydroSHEDS data) and the control of computing resource as resolution of atmospheric forcing data changes.

It is worth noting that the evaluation is carried out over 12 small to medium size rivers with different climatic and watershed
characteristics. Previous studies tended to focus on very large size rivers or one specific medium size catchment (e.g. Gong et al., 2009). This limits the generic application of their RRS at the global scale and may require parameter recalibration for smaller scale implementation. The role of calibration for realistic simulating runoff is discussed in Ducharne et al. (2003) and Beck et al. (2016). However, at this stage of our study, the good representation of complex basin maps provides a satisfactory basis for simulations. The fact that the new RRS explicitly accounts for the higher quality topographical information from
HydroSHEDS probably compensates the disadvantage of using simple reservoir parameters for all rivers, which is a legacy from the old RRS. Besides, the impact of uncertainty in forcing data on the simulated discharge is again emphasized. This is consistent with previous studies highlighting the necessity of accurate precipitation inputs for calculating water balance (e.g. Fekete et al., 2004; Decharme and Douville, 2006b). For the Mediterranean region we could also note that when the deviations from observations where larger than the forcing uncertainty, human management of water played an important role in the basin.
This suggests that anthropogenic processes are probably more critical in this region than our limits on representing geophysical processes.

Despite these uncertainties in the precipitation forcing data, the new RRS is shown to satisfactorily reproduce the seasonal variations of streamflow in a large range of catchments. The results presented in this study also show that the new RSS has the potential to reproduce adequately streamflow at daily timescales, which is not an easy task in LSMs. Balsamo et al. (2011)
showed preliminary river discharge predictions at daily timescales with the ECMWF land surface scheme. Only moderate correlation of 0.33 over 211 selected basins were found which is lower than the values presented here, so the new RRS presented in the present study provides a powerful tool for good representation of river basins in the ORCHIDEE LSM. Reproducing river runoff at daily frequency in global hydrology model often requires parameterization of complicated processes (Yamazaki et al., 2011; Beck et al., 2016). Further improvements could be expected by describing how stream velocity increases with
streamflow, which is important for simulating short time scale fluctuation of river discharge (Arora and Boer, 1999; Ngo-Duc et al., 2007b). In order to calculate the flow velocity, river width could be classically obtained using geomorphological relationships with annual mean river discharge (e.g. Leopold and Maddock, 1953), but it is also directly available based on remote-sensing (Yamazaki et al., 2014; Allen and Pavelsky, 2015). Above all, this study focuses on small river basins in complex topography as those flowing into the Mediterranean basin and there is a need to verify these results at the global scale
and on larger basins. A number of hydrological processes are still neglected in the current version of the new RRS. Thus, after




having demonstrated the numerical robustness of the RRS, we will have the possibility to improve the scheme by adding these missing processes, such as variable flow velocity, groundwater-river interactions, or river damming, whether for irrigation, hydropower generation, or navigation.

## 7   Conclusions

This study presents an attempt to revise the river routing scheme in the ORCHIDEE LSM, in order to benefit from the accurate hydrography from the HydroSHEDS database at 1 km resolution. This high resolution information is aggregated in Hydrological Transfer Units which are constructed inside each ORCHIDEE grid box. River networks which are depicted by connecting these HTUs provide precise flow pathways. The results show a wide range of river catchment size can be delineated precisely by the new RRS. The routing technique which is still based on a simple the linear reservoirs which requires information on the
orography and slopes. This information is improved by the accurate altitude provided by the HydroSHEDS data. In addition to satisfactory simulations at monthly timescale, the new RRS promises the ability to capture well other frequencies in the surface freshwater flows.

In order to improve the simulations of the current RRS, future work needs to concentrate on few aspects. Firstly, it is recommended that water management through reservoirs and abstraction should be integrated in the new RRS. Addressing the
interaction of human activities and natural river systems in the ORCHIDEE model is a necessary issue. The deficit of river discharge in summer time for irrigation is not captured in the new RRS.Another example is the Nile river, which could not be included in this study, as it is strongly affected by the association of irrigation and dam operation. Yet, the reduction of the fresh water flux from the Nile river has a significant impact on the thermohaline circulation of the Mediterranean Sea (Skliris and Lascaratos, 2004). Secondly, another interesting development path concerns the interactions between the groundwater system
and the rivers. Pappenberger et al. (2010) highlights that the groundwater delay parameter is the most sensitive calibration parameter in routing schemes, and a more physically based description of this parameter is being examined for the ORCHIDEE RRS (Schneider, 2017). Finally, hydraulic processes such as water storage in floodplains, swamps should be represented. Their importance in a global river routing model is underlined in Ducharne et al. (2003), Yamazaki et al. (2011) and Guimberteau et al. (2012a), and may benefit from the composite wetland map recently proposed by Tootchi et al. (2018). The RRS also lacks
a special treatment for lakes, which could be included based on the ideas of Milly et al. (2014). In the current version, water which flows to lakes will be evaporated through the soil moisture module of the ORCHIDEE model. Eventually, this RRS and its planned improvements will enhance the Earth system model by producing more realistic riverine freshwater flux into the Mediterranean Sea and improving the coupling between land and ocean processes.

## 8   Code availability

The source code is freely available online via the following address: http://forge.ipsl.jussieu.fr/orchidee/wiki/GroupActivities/ CodeAvalaibilityPublication/ORCHIDEE_gmd-2018-57. A DOI has been requested for this page which provides guidelines





for downloading. Readers interested in running the model should follow the instructions at http://orchidee.ipsl.fr/index.php/

5   you-orchidee.

*Acknowledgements.* The authors acknowledge helpful advice from two anonymous reviewers. We thank Dr. Luca Brocca (CNR-IRPI, Italy) for providing us the daily discharge of station Pontelagoscuro and Roma. We also gratefully acknowledge the GRDC (Global Runoff Data Centre) for providing valuable data. This work was supported by computing resource of the IPSL ClimServ cluster at École Polytechnique, France.



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





**Figure 2.** A schematic diagram of catchment construction in the revised river routing scheme of the ORCHIDEE land surface model (a) and illustration idea of Hydrological Transfer Unit (HTU): (b) Number of upstream HTUs contributing water to each grid box of the Rhône River at resolution of 1/8° with a maximum number of HTUs per grid box of 9. Color is given in logarithm scale. Red rivers come from the Generic Mapping Tools dataset (http://gmt.soest.hawaii.edu). The small orange box shows the grid box of (c) and (d). (c) The orange ORCHIDEE grid box in (b) with information derived from HydroSHEDS at the 30 arc-second resolution: arrows show the flow direction of each HydroSHEDS pixel; brown to blue colors show flow accumulation; bold red arrows show the flow direction of the grid box outlets; the orange lines delineate the boundary of the preliminary HTUs; and the orange circle highlights the outlet points of the largest preliminary HTU in this grid box. (d) Partitioning of HTUs which share the same grid box outlet (marked by the orange circle). Hexagons denote the outlets of inter-HTUs based on the Pfafstetter codification and shaded color indicate the different HTUs. Flow direction arrows are displayed in black and white. Orange squares highlight the approximate 1 $km^2$ HTUs.



**Figure 3.** Comparison of modeled area (using the old and new RRS) and reference area (from Wu et al. (2012)) for 12 river basins in this study (a). Representation of the Tiber River basin in the ORCHIDEE model with the old (b) and new (c) RRS at a regular latitude-longitude grid of 1/2°; and with the new RRS at grid of 1/16°(c). Colors show upstream area ($km^2$) contributing water to each grid box. The orange circle is the outlet point. Blue rivers come from the Generic Mapping Tools dataset (http://gmt.soest.hawaii.edu).





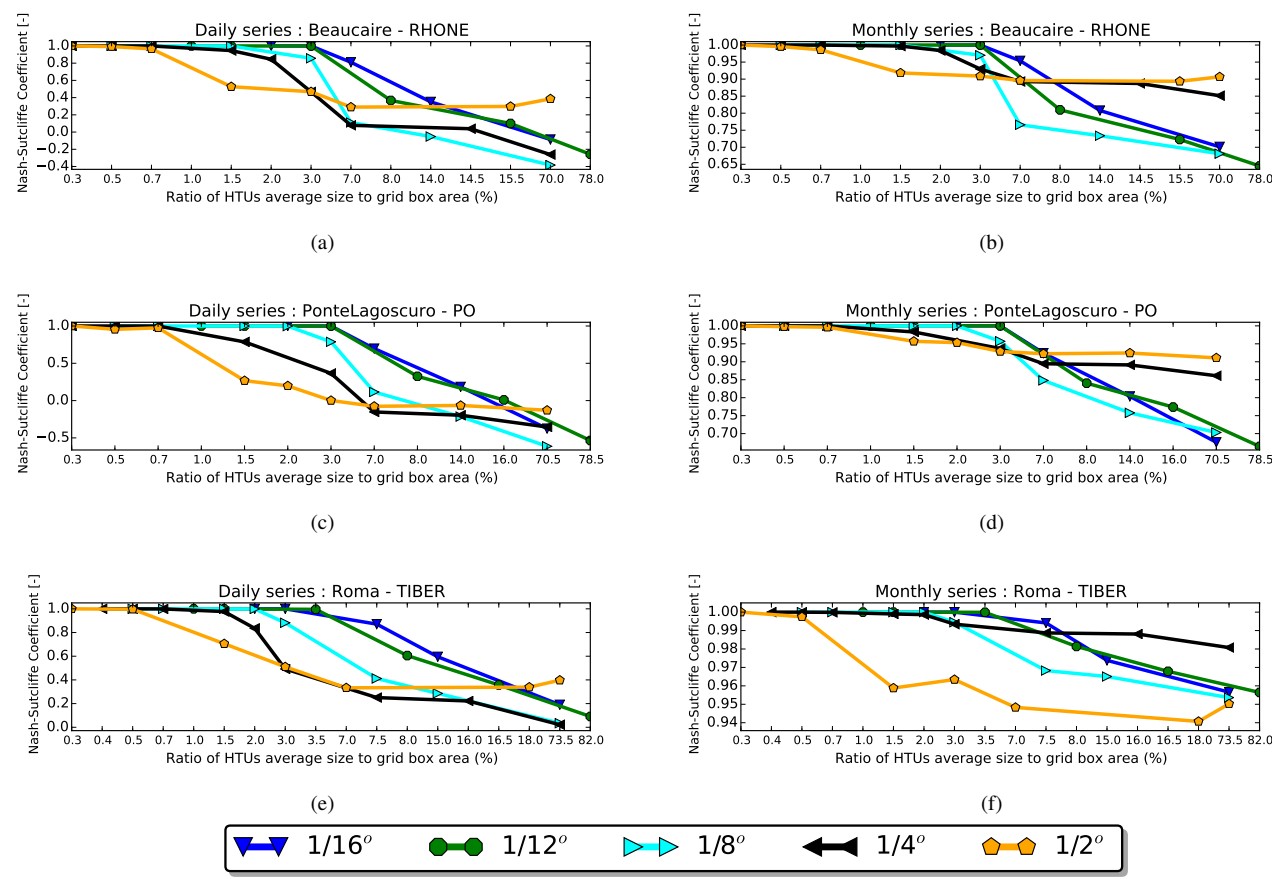

**Figure 4.** Sensitivity of simulation results (Nash-Sutcliffe coefficients) to different HTU sizes (HTU) in the ORCHIDEE mesh, with spatial resolutions varying from $1/16°$ to $1/2°$. The Nash-Sutcliffe coefficient is calculated with respect to the reference case (see the text for details) at daily (left column) and monthly (right column) timescales. Note that the range of Nash-Sutcliffe coefficient is different for each panel. The x-axis gives the ratio of the average HTU size to the ORCHIDEE grid box area (in %). The top row is station Beaucaire (the Rhône River), the middle is station Pontelagoscuro (the Po River) and the bottom is station Roma (the Tiber River).

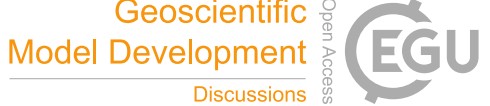

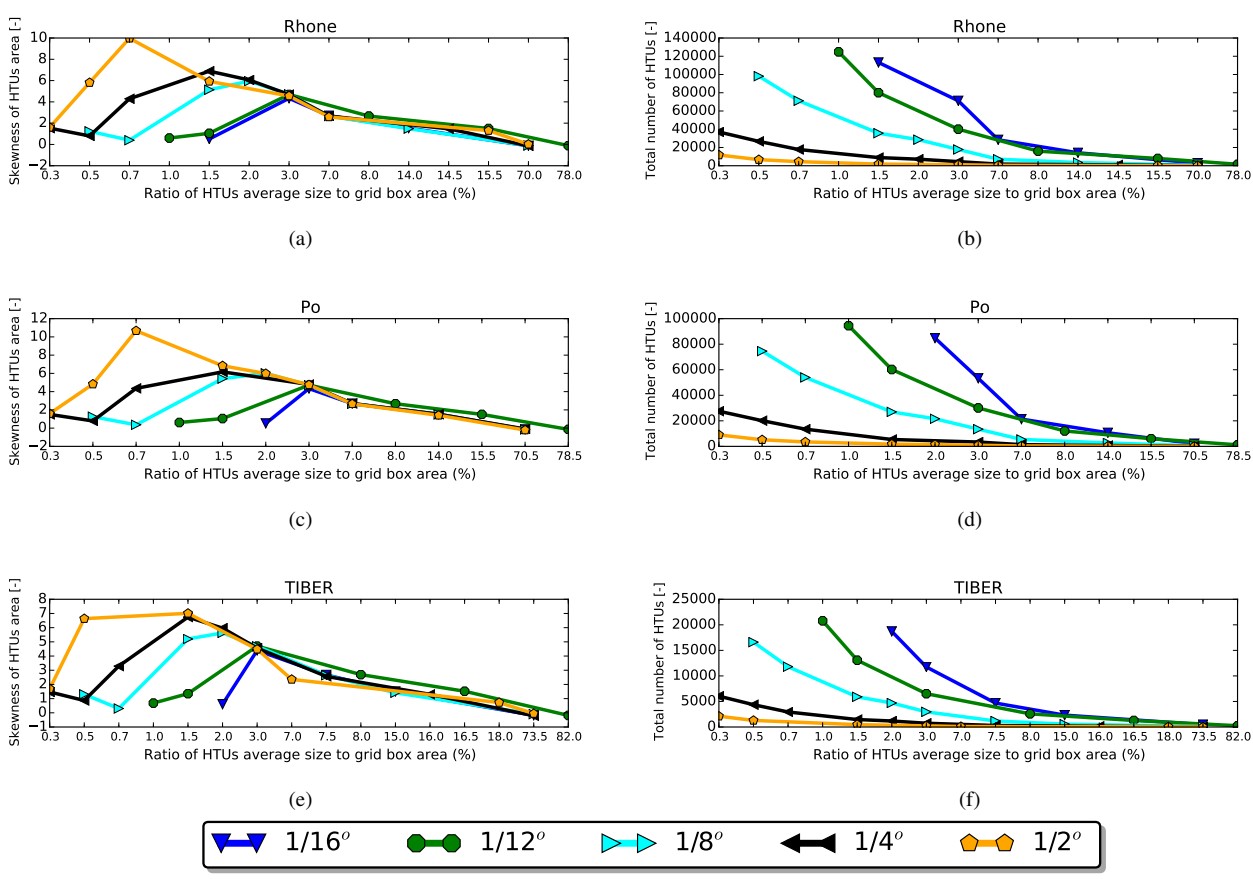

**Figure 5.** Same sensitivity analysis as in Figure 4, but for the skewness of HTUs area distribution (left column) and the total number of HTUs (right column).





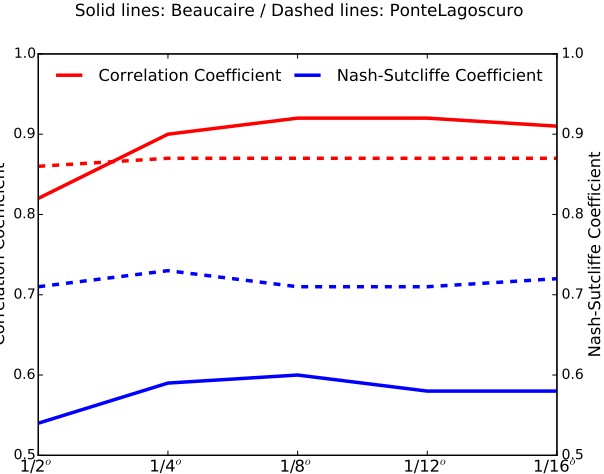

**Figure 6.** Performance of the new RRS as the resolution of the ORCHIDEE mesh changes from $1/2°$ to $1/16°$. All simulations are performed with the same average HTU area of about 13 km$^2$. Their performance is assessed at the monthly timescale against observed monthly time series at stations Beaucaire (solid lines) and Pontelagoscuro (dashed lines). Red lines are for the correlation coefficient and blue lines for the Nash–Sutcliffe coefficient.





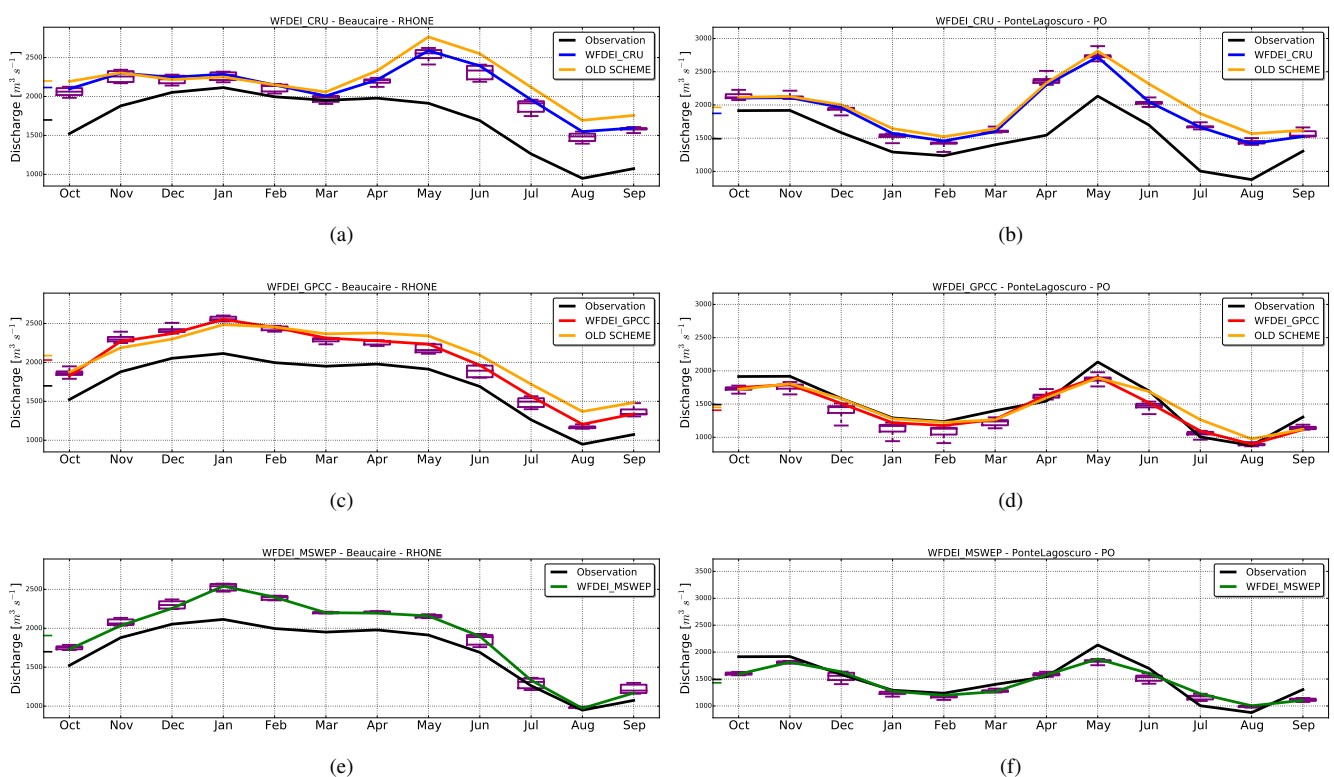

**Figure 7.** Mean annual cycle of river discharge at stations Beaucaire (left column) and Pontelagoscuro (right column) simulated with 3 different forcing data (WFDEI_CRU: top, WFDEI_GPCC: middle and WFDEI_MSWEP: bottom). The solid black line shows the observed discharge, and the orange line shows the simulation results with the old RRS. The other lines show simulation results with the new RRS and the practical average HTU size (see text for detail). The small colored ticks along the y-axis give the average values, and the purple boxplots show the range of results from simulations with the new RRS and different average HTU sizes (the limits of the boxes correspond to the first and third quartiles).



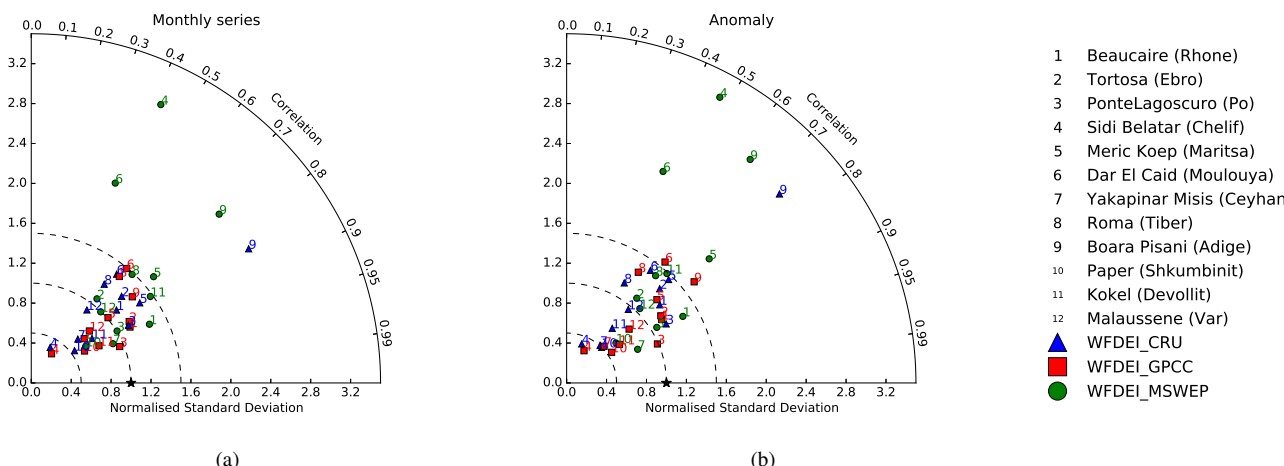

**Figure 8.** Taylor diagrams comparing simulated and observed river discharge simulation at 12 stations for (a) monthly series and (b) monthly anomalies with respect to the mean seasonal cycle. Blue triangle, red square and green circle correspond to simulations with forcing data from WFDEI_CRU, WFDEI_GPCC, WFDEI_MSWEP, respectively. The number of each station is given in the figure legend and in Table ListofStation. The black star shows the observed value at which normalized standard deviation and correlation coefficient are 1.0.

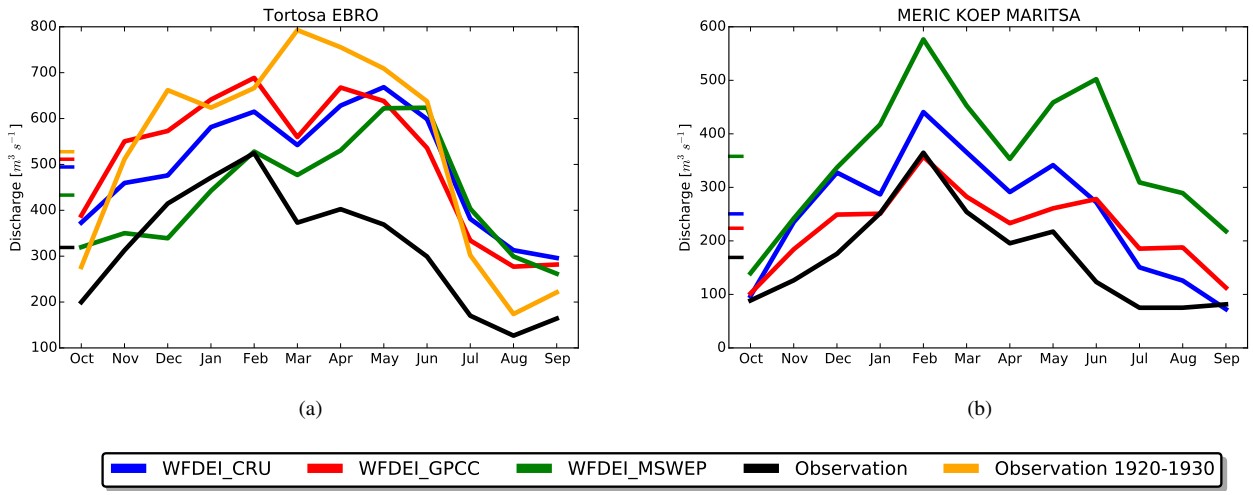

**Figure 9.** Mean nnual cycle of river discharge at station Tortosa on the Ebro river (a) and Meric Koep on the Maritsa river (b) simulated with forcing data from WFDEI_CRU (blue), WFDEI_GPCC (red) and WFDEI_MSWEP (green). The solid black line is the observation data after 1979, while the orange line is the observation data for period 1920-1930 (at station Tortosa). The small colored ticks along the y-axis give the average values.



**Figure 10.** Flow duration curve for daily river discharge at station Beaucaire (a), Pontelagoscuro (b), Yakapinar Misis (c) and Sidi Belatar (d). The solid black line is the observation data. The blue, red and green are simulations with forcing data from WFDEI_CRU, WFDEI_GPCC and WFDEI_MSWEP, respectively.

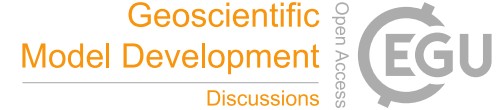





(a) River discharge

(b) Sub-surface runoff

(c) Precipitation

(d) Evaporation

**Figure 11.** Power spectral density of daily river discharge (a) estimated by the ORCHIDEE at the station Beaucaire (Rhône) and sub-surface runoff (b), precipitation (c), evaporation (d) over entire Rhône river catchment estimated by the ORCHIDEE. The Savitzky-Golay filter with window of 21 is applied for smoothing noise signal. The black line is the observation data (Obs) and the green is the ORCHIDEE model results with forcing data from WFDEI_MSWEP (ORCHIDEE) Trend lines for frequency above/below 30 days are shown with their $\beta$ slope.