# Peer review of "ORCHIDEE-ROUTING: Revising the river routing scheme to use a high resolution hydrological database"

_Geoscientific Model Development, 2018_

## Short Comment (SC1) · 14 May 2018

The access to this version of ORCHIDEE is now available through a DOI :

http://dx.doi.org/10.14768/06337394-73A9-407C-9997-0E380DAC5593

Jan Polcher

---

## Referee Comment (RC1) · L.A. Melsen (Referee) · 17 May 2018

The manuscript of Nguyen-Quang et al. discusses a new, high resolution routing scheme for the ORCHIDEE model. Although the concept of the routing has not been adapted compared to an earlier version of ORCHIDEE routing, the spatial resolution has been increased significantly to better represent river basin boundaries and the flow paths within the basin. The manuscript is generally well written (some textual revision needed here and there) with informative illustrative figures. Overall, I think the manuscript would be suitable for publication in GMD. There are, however, a few topics and points which I would like to see discussed in somewhat more detail.

[Figure]

Even though not a classical manuscript-structure was followed, I appreciated the current set-up in which Section 4 describes HTU-sensitivity (with a nice and clear conclusion in Sect. 4.3) and Section 5 the performance compared to observations.

Major

The manuscript clearly demonstrates the uncertainty introduced by forcing data. Furthermore, uncertainty introduced by human impact, and model structural uncertainty in ORCHIDEE have been discussed. Also uncertainty in the concept of the routing scheme are discussed (does not account for stream flow velocity changes, bankoverflow, etc.). As a reader, it is still unclear to me why the priority was set to improving the spatial resolution of the routing, rather than to any of these other sources of uncertainty (which are substantial). The results clearly demonstrate that the improvements from the higher resolution routing model are difficult to validate given all the other sources of uncertainty.

Continuing on this point, p.3 lines 3-5 refer to the discussion of global hyper-resolution models. This fits well in line with what has been done in this paper, but overlooks the discussion that is going on within hydrological sciences on whether focussing only on increasing the spatial resolution is the best way forward (see e.g. the comment that Beven and Cloke wrote).

In Section 3.4, three reservoir parameters have been calibrated based on the Rhone data, and subsequently these parameters have simply been applied to all basins. That does make me wonder how sensitive the routing model is to these parameters, and how different the parameters are when calibrated on another basin. I think this is too important to simply step over like is done now.

Many results are presented at the monthly time scale. First thing I wondered is the added value of the high resolution routing for the monthly timescale, because in principle all that routing does is delaying the water over time, but given the size of the basins, this effect is expected to be limited on the monthly time scale. So, then the added value

of the improved routing is basically only in better delineation of the basins compared to the coarse resolution scheme, for which a complex DEM and flow direction scheme is not necessary. Could you respond to that?

The analysis on the daily time scale (e.g. Sect 5.3) to discuss model performance seems therefore more relevant, but only shows results in terms of a flow duration curve, and from a flow duration curve it is hard to retrieve timing-information. Is it not possible to include hydrograph-information to demonstrate the simulations versus observations directly?

The current conclusion (sect 7) consists of one paragraph with the results, while the second paragraph is basically only recommendations. I think it would improve the strength of the paper to not end with mentioning everything that is missing still, but describe what has been added with this rrs. Furthermore, the results of section 4 and 5 could be touched upon in the conclusion.

Minor

General; there is a tendency throughout the manuscript to describe the figure legend in the main text (see e.g. p.8 l.15). This is not necessary and makes the manuscript less nice to read. Describe the legend in the figure caption and the conclusions from the figure in the main text.

As not being very familiar with the models discussed in the Introduction, I don't see the logic between the section starting on p.2 l.7 and the section on p.2 l.20. The first section says: ".. redistribution in turn can feed other processes in the LSM (floodplain evaporation, irrigation)." Then it continues to describe how post-processing routing schemes neglects feedback interactions between river discharge and soil hydrology. In the second section, the ORCHIDEE routing is described, but as far as I could find in the manuscript, this routing also does not account for floodplain evaporation, irrigation, or river discharge / soil hydrology interaction. Therefore this is confusing to the reader.

Figure 1: the red colour appears in the legend, but is also used to indicate the regions of this study. It is unclear if the red colour then still refers to the legend colour too.

p.6 l.7: it is unclear how this scheme allows for irrigation withdrawal and flood plains. Perhaps elaborate on that. Furthermore, it is correct that irrigation is not accounted for in this study, right? If the routing allows for that, why was it excluded in a application in a region where irrigation is expect to play an important role?

p.14 l. 28 I am not familiar with the power spectrums discussed here. Perhaps some more information on this methodology can be provided, and what the implications of the results are.

p.17 l. 15 I don't see how the results support this (strong) statement.

References

Beven, K. and Cloke, H.: Comment on "Hyperresolution global land surface modeling: Meeting a grand challenge for monitoring Earth's terrestrial water" by Eric F. Wood et al., Water Resour. Res., 48, W01801, doi:10.1029/2011WR010982, 2012.

---

## Referee Comment (RC2) · Anonymous Referee #2 · 25 May 2018

Summary: The paper presents the implementation of the river network derived from high resolution topographic data to the river routing model in ORCHIDEE land surface model and its effects on the streamflow estimations compared to the previous coarser resolution network data. The paper discusses the optimal sub-basin sizes (called HTUs here), and also presents the simulation skills at the 12 Mediterranean basins.

Recommendation: The manuscript contains interesting analyses, but I think the paper can be improved. I would suggest the authors go through major revision.

General Comments:

1. Regarding the paper readability, the paper can be improved by making the de-

scriptions more concise throughout the paper, but especially section 3 and discussion. Section 3.3. section seems to be important, but it is a little lengthy, therefore, it is hard to follow. I would like to suggest describing each step in Figure 2a) one by one at the beginning. Also note the specific comments.

2. The paper emphasizes an importance of incorporating anthropogenic processes on river flow (irrigation, reservoir etc.) at multiple locations throughout the paper (P3-L7, P6-L5-8, P17-L20, P18-L13-15). I agree that this is one of future direction for river modeling. However, this issue is not main topic of this paper, and the paper never mentioned how high resolution gridded network-based model helps to incorporate this nontrivial processes as indicated in P3-L7. I would like to suggest at least how high resolution river network routing helps incorporate these processes

3. The paper states that the authors revised the river routing scheme. This gave me an impression that the author revised actual river routing algorithms, but the paper's contribution is to develop the method to derive river network and basin delineation and routing parameters based on high resolution DEM. Also, I understood that the routing scheme (unit-to-unit routing scheme) has not changed from the previous model. I would suggest describing title more specifically and also reword the routing scheme in the text to specify precisely what is really done in the paper (e.g., P1-L2, P1-L9. P6-L10.).

4. Improvement of streamflow estimates, in particular, volume bias, are due to the better representation of sub-basin areas than the previous model. I believe this, but to illustrate this clearly, I would suggest including total runoff depth over the sub-basin based on the new (and old) river network and compare it with observed runoff depth. This can be added in Figure 3.

5. The paper discusses some issue on computational cost on high resolution river network data (Section 4.3). For GMD paper, I think this is very appropriate. routing model is much less computationally expensive. The paper state recommended HTU

resolution improve the computational cost by a factor of 10 compared to the highest (native DEM) resolution in P11-L21. Is this factor scaled for the river basin area? I would also like to suggest stating quantified computation cost (e.g., wall-clock, or core-hours = number of processor cores x wall-clock in hours). a factor of 10 might not be significant compared to the LSM computational cost.

Specific comments

P2, L11. I am not sure why scaling causes information loss. I feel I disagree with this statement, but I guess I did not understand the statement.

P3, L6-7. This is naïve statement to me. Do you mean this by hyper-resolution LSM? Hyper-resolution LSM model does not necessarily provide better simulations (if forcing is not good). Hyper-resolution LSM may provide accurate representation of some of geophysical properties (e.g., topography), but not likely soil, geology.

P7, L8. "The area of each HTU is limited by an user-defined size". This is not crystal-clear to me. I understood the "maximum" area of each HTU is set by user to constrain the HTU areas. Correct?

P7, L27. I thought there were eight smaller HTUs and 4 inter-basins. I wonder if I misunderstood something.

P8, L7-10. Regarding combining the small HTUs, how do you determine the outlet of the combined HTUs.

P9, L13-14. I am having difficulty in understanding this sentence (how this derivation of k corresponds to unit hydrograph).

P9, L15. I am also having difficulty in this sentence and equation (LHS looks like the velocity over the HTU, and RHS looks induvial 1km pixel velocity?). Also what is D?

Last paragraph in Section 3.3. Most of sentences in this paragraph seems to be out of place (maybe fit in introduction). I would suggests removing to make this section

shorter.

P17, L22-27. Performance of streamflow simulation is based on all the model components (including forcing). These sentences weigh too much on RRS to attribute the simulation performance or uncertainty.

I would suggest combining discussions and conclusions potentially sub-section (e.g., limitation of model, future work etc.). The first paragraph in discussion sounds like reading introduction. I would suggest removing (moving to introduction) or making it very short.

P18, L9-10. This sentence should appear earlier (section 3.1?)

P18, L14-19. Issues on Nile river is already mentioned earlier, and this sentence seems to be out of place. Suggest removing this.

Table 3. Why do some rivers have two rows? For example, what is 6.83 and 5.99 for NS of Boara Pisani?

---

## Author Comment (AC1) · 6 Jul 2018

**Answers to the interactive comments from the anonymous referee #2**

Firstly, the authors would like to thank you the careful review from the anonymous referee #2. Please allow us to answer below the comments which you gave in the interactive comments. Please note that the **RC** stands for the referee comments, **AR** is the abbreviation of author's response and **AC** indicates the author's changes in manuscript.

For the general comments:

1. More concise descriptions

   **RC:** Regarding the paper readability, the paper can be improved by making the descriptions more concise throughout the paper, but especially section 3 and discussion. Section 3.3. section seems to be important, but it is a little lengthy, therefore, it is hard to follow. I would like to suggest describing each step in Figure 2a) one by one at the beginning. Also note the specific comments.

   **AR:** *The authors will refine the text for more concise descriptions with the focus on the section 3 and discussion.*

   **AC:** *The modification can be found in p.7 l.20, p.8 l.11.*

2. The anthropogenic processes

   **RC:** The paper emphasizes an importance of incorporating anthropogenic processes on river flow (irrigation, reservoir etc.) at multiple locations throughout the paper (P3-L7, P6-L5-8, P17-L20, P18-L13-15). I agree that this is one of future direction for river modeling. However, this issue is not main topic of this paper, and the paper never mentioned how high resolution gridded network-based model helps to incorporate this nontrivial processes as indicated in P3-L7. I would like to suggest at least how high resolution river network routing helps incorporate these processes.

   **AR:** *The authors will provide a perspective of how high resolution river network can assist the incorporation of irrigation. In fact, in the earlier version of the routing scheme, the representations of irrigation and floodplain already existed (De Rosnay et al., 2003; Guimberteau et al., 2012b; d'Orgeval et al., 2008; Guimberteau et al., 2012a). Since these representations of human processes or flood plains is based on a hypothesis that HTU were rather large (i.e. at scale of $1/2^o$), this is not valid any more and thus these parameterizations need to be revised in order to work with high resolution descriptions of the river basins. In order to represent the irrigation processes in the ORCHIDEE model, the irrigated area map is required to integrate into the routing scheme. The most update global map of the irrigated area is provided at resolution of approximate $10 \ km$ by Siebert et al. (2013). The new routing scheme constructs the river networks by connecting the HTUs which can be vary in size from about $1 \ km^2$ to the area of the ORCHIDEE grid cell (e.g. $1/2 \ ^o$). Hence, the irrigated location can be located in each HTUs. The difficult part is how to control the store and release processes of the water along the river network that is irrigated. For example, a irrigated region can receive water from different tributaries of a river network as well as depend on the available upstream water.*

   **AC:** *In the Discussion, the authors suggest how the high resolution network can assist the integration of irrigation in the routing scheme based on the applied idea in the earlier version. Considering all the comments of the reviewer, the arguments of the manuscript is re-organized to make it clear that the old parameterizations cannot work at high resolution and need to be revised. Please also consider the author's reply to the first reviewer which is raised from the same issue.*

3. The title of the article

   **RC:** The paper states that the authors revised the river routing scheme. This gave me an impression that the author revised actual river routing algorithms, but the paper's contribution is to develop the method to derive river network and basin delineation and routing parameters based on high resolution DEM. Also, I understood that the routing scheme (unit-to-unit routing scheme) has not changed from the previous model. I would suggest describing title more specifically and also reword the routing scheme in the text to specify precisely what is really done in the paper (e.g., P1-L2, P1-L9. P6-L10.).

   **AR:** *The authors totally agree with the reviewer. The authors will correct the title and reword the mentioned parts.*

**AC:** *The title is changed as: "ORCHIDEE-ROUTING: Revising the river routing scheme using a high resolution hydrological database". The Abstract and Introduction are reworded to precisely describe the study in this paper.*

4. Total runoff depth

**RC:** *Improvement of streamflow estimates, in particular, volume bias, are due to the better representation of sub-basin areas than the previous model. I believe this, but to illustrate this clearly, I would suggest including total runoff depth over the sub-basin based on the new (and old) river network and compare it with observed runoff depth. This can be added in Figure 3.*

**AR:** *Runoff depth means that river discharge get divided by the upstream area. In our case, the up-stream area does not change more than 1% from one HTU resolution to the other, so the runoff depth does not change.*

**AC:** *No modification in the text.*

5. The computational time

**RC:** *The paper discusses some issue on computational cost on high resolution river network data (Section 4.3). For GMD paper, I think this is very appropriate. routing model is much less computationally expensive. The paper state recommended HTU resolution improve the computational cost by a factor of 10 compared to the highest (native DEM) resolution in P11-L21. Is this factor scaled for the river basin area? I would also like to suggest stating quantified computation cost (e.g., wall-clock, or core- hours = number of processor cores x wall-clock in hours). a factor of 10 might not be significant compared to the LSM computational cost.*

**AR:** *The authors totally agree that the computational cost is a worthy discussion point in a GMD paper. For the resolution of $1/2^o$ of the ORCHIDEE model, the simulation time for the entire simulation domain presented in the paper can be reduced from about 30 hours to more than 3 hours. This is based on the wall-clock check in the ClimServ server (http://climserv.ipsl.polytechnique.fr/)*

**AC:** *More information about computational cost is added after p.11 l.21.*

For the specific comments:

1. An argument in the Introduction

**RC:** P2, L11. I am not sure why scaling causes information loss. I feel I disagree with this statement, but I guess I did not understand the statement.

**AR:** *The argument is: "Of course, this transfer ignores feedback interactions between river discharge and soil hydrology of the LSM. It also accepts the loss of information as aggregating different spatial and temporal discretization between two models." The authors will refine this statement to express the idea more clear. Normally, the LSM and the standalone routing scheme run on different grids. As the exercise of routing flow is bound by topographic details, a higher resolutions is desirable. This means that the flux of water from the LSM to the routing scheme has to be extrapolated. On the other side, if transported water needs to return to the LSM because it goes back into evaporation or soil moisture, it needs to be aggregated. Interactions between the vertical movement of water (represented by the LSM) and horizontal movements (river routing) becomes more complicated and are filtered by the extrapolation/aggregation.*

**AC:** *The sentence is refined as: "If aggregating different spatial and temporal discretization between two models requires interpolation step, it can cause the loss of information."*

2. The statement about hyper-resolution LSM

**RC:** P3, L6-7. This is naïve statement to me. Do you mean this by hyper-resolution LSM? Hyper-resolution LSM model does not necessarily provide better simulations (if forcing is not good). Hyper-resolution LSM may provide accurate representation of some of geophysical properties (e.g., topography), but not likely soil, geology.

**AR:** *The statement is: "In particular, a hyper-resolution model allows providing more precise fresh water fluxes for ocean circulation simulation." The authors will remove this statement.*

**AC:** *The sentence is removed from the text.*

3. The maximum area of each HTU

   **RC:** P7, L8. "The area of each HTU is limited by an user-defined size". This is not crystal- clear to me. I understood the "maximum" area of each HTU is set by user to constrain the HTU areas. Correct?

   **AR:** *The user can choose the average size of the HTU in relation to the grid size. This allows to use Figure 4 of the graphic to select an average HTU size which does not degrade too much the simulated discharge. The authors will refine this sentence.*

   **AC:** *The phrase is modified as: "The user need to define the average area of HTUs in each ORCHIDEE grid cell."*

4. The Pfafstetter topological coding system

   **RC:** P7, L27. I thought there were eight smaller HTUs and 4 inter-basins. I wonder if I misunderstood something.

   **AR:** *The entire sentence is: "The partitioning process relies on the Pfafstetter topological coding system for streams and basins (Pfafstetter, 1989; Verdin and Verdin, 1999): the flow accumulation is used to identify the main stream of the HTU to partition, and its main four tributaries; this results in dividing the large HTU into nine smaller HTUs comprising the basins of the four tributaries and five inter-basins." This is the original idea of the Pfafstetter code. If you can find 4 tributaries which correspond to 4 crossed points with the main stream, there will be 5 inter-basins which split by these 4 crossed points. But there is sometimes only one crossed point for 2 main tributaries then there are only 3 crossed points and 4 inter-basins.*

   **AC:** *No modification in the text.*

5. The outlet of the combined HTUs

   **RC:** P8, L7-10. Regarding combining the small HTUs, how do you determine the outlet of the combined HTUs.

   **AR:** *Among all small HTUs which we want to combine, the two smallest HTUs will be combined first and the outlet of the new HTU will be the outlet of the combined HTUs. An iteration is made to combine HTUs until reaching the desired number of HTUs.*

   **AC:** *No modification in the text.*

6. A sentence and the equation in Section 3.4

   **RC:** P9, L13-14. I am having difficulty in understanding this sentence (how this derivation of k corresponds to unit hydrograph). P9, L15. I am also having difficulty in this sentence and equation (LHS looks like the velocity over the HTU, and RHS looks induvial 1km pixel velocity?). Also what is D?

   **AR:** *We will correct this part of the manuscript for better description of the water routing process. The argument using the idea of unit hydrograph is removed. Runoff is routed downstream with a delay time that is controlled by the number of HTUs along the stream, and the properties of each HTU, namely their slope index k and reservoir parameter g, the product of which defines the time lag of each HTU. The slope index is first calculated at the 1-km resolution based on the slope and length of the HydroSHEDS pixels (i.e. mentioned with a formula in Section 3.1). Then it is aggregated at the HTU scale by an algorithm which uses the drainage directions and the resulting distance of each pixel to the HTU outlet. For each pixel, we define K as the sum of all the 1-km values of k along the corresponding downstream line. The upscaled value of k for the HTU is then given by the product of the sum of K across all the pixels composing the HTU and the fractional area of the HTU. As a result, the slope index of HTUs changes with the area and length of stream lines in the HTUs, so that the streamflow velocity does not depend, or weakly, on the HTU scale.*

   **AC:** *After P.9 L.13, the text is modified.*

7. The last paragraph in Section 3.3

   **RC:** Last paragraph in Section 3.3. Most of sentences in this paragraph seems to be out of place (maybe fit in introduction). I would suggests removing to make this section shorter.

   **AR:** *The authors will remove this paragraph and separate the arguments to other section.*

   **AC:** *The last paragraph in Section 3.3 is removed.*

8. The discussion part

   **RC:** P17, L22-27. Performance of streamflow simulation is based on all the model components (including forcing). These sentences weigh too much on RRS to attribute the simulation performance or uncertainty. I would suggest combining discussions and conclusions potentially sub-section (e.g., limitation of model, future work etc.). The first paragraph in discussion sounds like reading introduction. I would suggest removing (moving to introduction) or making it very short.

   **AR:** *The authors will refine the first paragraph in discussion.*

   **AC:** *The first paragraph in discussion is made shorter and is combined with the second paragraph.*

9. The conclusion part

   **RC:** P18, L9-10. This sentence should appear earlier (section 3.1?)

   **AR:** *The sentence is: "The routing technique which is still based on a simple the linear reservoirs which requires information on the orography and slopes." The authors will remove this sentence and put the idea earlier.*

   **AC:** *Removed sentence. Considering also the comments from another reviewer, the conclusion section is re-organized.*

10. The Nile river issues

    **RC:** *P18, L14-19. Issues on Nile river is already mentioned earlier, and this sentence seems to be out of place. Suggest removing this.*

    **AR:** *The issues on Nile river is mentioned earlier in Section 2.1 with only one sentence to notice the exclusion of the Nile river from the simulation domain. The Nile issue is recalled in the Conclusion part to support the issue of necessity to address the interaction of human activities and natural river systems in the ORCHIDEE model. The Discussion and Conclusion are also rewritten hence this argument is presented more properly.*

    **AC:** Re-arrange this issue along with re-organize the Discussion and Conclusion part.

11. The error in the format of Table 3

    **RC:** Table 3. Why do some rivers have two rows? For example, what is 6.83 and 5.99 for NS of Boara Pisani?

    **AR:** *It's the mistake in the format of Table 3. The negative number is displayed in two rows. The authors will correct this format mistake.*

    **AC:** *Table 3 is re-formatted.*

**References**

De Rosnay, P., Polcher, J., Laval, K., and Sabre, M.: Integrated parameterization of irrigation in the land surface model ORCHIDEE. Validation over Indian Peninsula, Geophysical Research Letters, 30, 2003.

d'Orgeval, T., Polcher, J., and Rosnay, P. d.: Sensitivity of the West African hydrological cycle in ORCHIDEE to infiltration processes, Hydrology and Earth System Sciences, 12, 1387–1401, 2008.

Guimberteau, M., Drapeau, G., Ronchail, J., Sultan, B., Polcher, J., Martinez, J.-M., Prigent, C., Guyot, J.-L., Cochonneau, G., Villar, J. C. E., et al.: Discharge simulation in the sub-basins of the Amazon using ORCHIDEE forced by new datasets, Hydrology and Earth System Sciences, 16, 11 171–11 232, 2012a.

Guimberteau, M., Laval, K., Perrier, A., and Polcher, J.: Global effect of irrigation and its impact on the onset of the Indian summer monsoon, Climate dynamics, 39, 1329–1348, 2012b.

Siebert, S., Henrich, V., Frenken, K., and Burke, J.: Update of the digital global map of irrigation areas to version 5, Rheinische Friedrich-Wilhelms-Universität, Bonn, Germany and Food and Agriculture Organization of the United Nations, Rome, Italy, 2013.

---

## Author Comment (AC2) · 7 Jul 2018

Dear Dr. L.A. Melsen (the reviewer), Dear Dr. Chiel van Heerwaarden (the editor),

Thank you so much for your reading of my submitted article and all your thoughtful comments. I really appreciate your comments and advice. Please find my replies in the supplement file (in pdf format). I am really sorry that I still can not provide the revised manuscript today. I sent an email to Dr. Chiel van Heerwaarden (the handling topical editor) to explain my difficulty. I hope that the GMD journal will allow me to have an extended submission deadline for the revised manuscript of my article. It's really important for me. Thank you so much for your help.

[Figure]

Best regards, Trung Nguyen

Please also note the supplement to this comment:
https://www.geosci-model-dev-discuss.net/gmd-2018-57/gmd-2018-57-AC2-
supplement.pdf

————————————————

---

## Author Comment (AC3) · 9 Jul 2018

Dear Dr. Chiel van Heerwaarden (the editor), Dear Dr. L.A. Melsen (the reviewer #1), Dear the anonymous referee #2,

Thank you so much for all your thoughtful comments. I really appreciate your comments and advice. I uploaded my replies as the supplement files (in pdf format). I need to work on the manuscript until September in order to provide you the revised version. I sent an email to Dr. Chiel van Heerwaarden (the handling topical editor) to explain my difficulty. I hope that you (the two reviewers) and the GMD journal will allow me to have an extended submission deadline for the revised manuscript of my article. It's really

important for me. Thank you so much for your help.

Best regards, Trung Nguyen

---

## Author Response (AR1)

**Answers to the interactive comments from Dr. L.A. Melsen (lieke.melsen@wur.nl).**

Firstly, the authors would like to thank Dr. L.A. Melsen for the thoughtful review. Please allow us to answer below the topics and points which you figured out in the interactive comments. Please note that the **RC** stands for the referee comments, **AR** is the abbreviation of author's response and **AC** indicates the author's changes in manuscript. Page and line numbers indicated below belong to the marked-up manuscript version. There are 05 major points:

1. The priority to improve the spatial resolution of the routing scheme

   **RC:** The manuscript clearly demonstrates the uncertainty introduced by forcing data. Furthermore, uncertainty introduced by human impact, and model structural uncertainty in ORCHIDEE have been discussed. Also uncertainty in the concept of the routing scheme are discussed (does not account for stream flow velocity changes, bank overflow, etc.). As a reader, it is still unclear to me why the priority was set to improving the spatial resolution of the routing, rather than to any of these other sources of uncertainty (which are substantial). The results clearly demonstrate that the improvements from the higher resolution routing model are difficult to validate given all the other sources of uncertainty. Continuing on this point, p.3 lines 3-5 refer to the discussion of global hyper-resolution models. This fits well in line with what has been done in this paper, but overlooks the discussion that is going on within hydrological sciences on whether focusing only on increasing the spatial resolution is the best way forward (see e.g. the comment that Beven and Cloke wrote).

   **AR:** The improvement of the spatial resolution of the routing is the first and necessary step before any further developments of the model. It is an important step to improve the morphological description of river systems which is known to be a strong constraint on river flow. Indeed, we not only improve the basin areas but also the length and slope of rivers. It is important not only for describing human influences but also to physically describe river flow. Nevertheless, since the effect of all uncertainty sources (e.g. epistemic uncertainty) is difficult to separately evaluated, the spatial resolution might be a proper starting point to investigate. The more careful evaluation of the impact of these uncertainty sources on the operation of the river routing scheme is worthy another study. The new RRS provides the basis for further modification with a focus on the representation of human activities. In addition, only the descriptions of the river basins is improved, the structure of the model is kept the same as in the earlier version. This allows to investigate the importance of catchment delineation in the routing scheme. Modifications to the stream flow velocity, bank overflow, or human impacts can only made after the routing scheme operates at the new high resolution so that value of each improvement can be evaluated. About the argument of hyper-resolution models, the authors thank the reviewer for the reference from Beven and Cloke (2012), the authors will add more arguments on the part which discuss about the global hyper-resolution models.

   **AC:** Combining with other comments from two reviewers, the Introduction part is modified with the penultimate paragraph focuses on the reason of improving the spatial resolution of the RRS (p.2 l5-6, l12-16, l25, p.3 l14-19). About the argument of the hyper-resolution model, after the p.3 l.17, the authors added:"Certainly, refining grid scale cannot resolve the problems of epistemic uncertainties in hydrological predictions (Beven and Cloke, 2012), but it is an important step to improve the morphological description of river systems, which is a major driver of river flow."

2. The calibration of the three reservoir parameters

   **RC:** In Section 3.4, three reservoir parameters have been calibrated based on the Rhone data, and subsequently these parameters have simply been applied to all basins. That does make me wonder how sensitive the routing model is to these parameters, and how different the parameters are when calibrated on another basin. I think this is too important to simply step over like is done now.

   **AR:** Authors agree with the reviewer that the calibration of three reservoir parameters is an important issue, especially when the RRS is applied for one specific catchment. Indeed, two wonders of the reviewer are too complex and difficult to satisfactorily solved within the scope of this article. Figure 1 shows a simple experiment to answer the question of how sensitive the new RRS is to these parameters. The new RRS is run with new reservoir parameters (i.e. 0.01, 0.5 and 7.0 day/km) and old parameters (i.e. 0.24, 3, and 25 day/km) with maximum one HTU per a ORCHIDEE grid box. As can

be seen both in the mean annual cycle and monthly discharge anomaly series, there are not significant differences when applying two parameter sets. In this case, although the values of two parameter sets are quite different, we can see that the sensitivity to the reservoir parameters are not remarkable. One reason can be expected is that the slope index which calculated from the HydroSHEDS play an important role in the control of river flow. However, there is a caveat that only one HTU is constructed in each ORCHIDEE grid box. This threshold leads to a small number of HTUs in simulated river network for the Rhone (47 HTUs), the Po (48 HTUs) and the Tiber river (6 HTUs). Since being a unit-to-unit routing scheme, the new RRS is also sensitive to the number of HTUs. In fact, the impact of the number of HTUs and routing time step on the simulated river discharge cannot be separated from the effect of reservoir parameters.

In order to answer the second wonder of the reviewer of how different the parameters are when calibrated on another basin, figure 2 and 3 are shown. In application of the RRS, users can adjust these parameters to reproduce reasonably the historical observed hydrograph of river discharge at their study catchment. However, in this study, we focus on the application of new topography information from the HydroSHEDS data in order to better represent the topographical control on the stream flow speed at a a larger scale and independently of specific issue of individual catchments. The major idea is finding an acceptable parameter set for generic application worldwide. Therefore, the three parameters are calibrated at the station Beaucaire (the Rhone river) with the target of simulating river discharge comparably to the former version of the RRS. Figure 2 presents the sensitivity of the new RRS to the reservoir parameters in a theoretical test at the station Beaucaire (the Rhône river) and the station Pontelagoscuro (the Po river). In this test, the runoff and drainage are not gathered from the soil moisture module of the ORCHIDEE model. During the first 03 days, one $kg.m^{-2}.day^{-1}$ divided up in the runoff and drainage for all grid cells and the analyses are for daily time series of one year. The old routing scheme is used to generate reference case at the Rhône river due to its efficient applications in previous studies (Ngo-Duc et al., 2005b, a; De Rosnay et al., 2003; Guimberteau et al., 2012b, a). The core idea is to determine new reservoir parameter set that can reproduce the same simulated results as the old routing scheme. Both routing schemes are run at the horizontal resolution of $1/2^o$. In the old routing scheme, the Rhône river basin covers 65 grid cells of the ORCHIDEE model while it is only 48 grid cells in the new routing scheme. The new routing scheme is configured with parameter of stream reservoir ($g_1$) varied from 0.001 to 1.0 $day/km$, the overland reservoir parameter ($g_2$) is from 0.1 to 14.0 $day/km$ and the groundwater reservoir parameter ($g_3$) is from 1.0 to 60.0 $day/km$. The higher value of correlation coefficient shown in the Figure 2 (red color) highlights the parameter sets at which the new routing scheme reproduces results closed to the old routing scheme. As can be seen, the combination effect of three parameters on the simulation results are different between two rivers (i.e. CC values are scattered in different pattern). Determining a specific parameter set also requires the evaluation based on other evaluated metrics such as the root mean square error and standard deviation as shown in the Taylor diagram of Figure 3. In this case, for the Po, the best choice for the parameter set can be 1.0, 3.0 and 7.0 day/km. While they can be 0.01, 0.5 and 25.0 day/km for the Rhone. The chosen parameter set in the article is, at some extents, a satisfactory solution both for the Po and Rhone. Above all, the sensitivity of the simulated discharge to these reservoir parameters requires a comprehensive study which considers more case studies. The theoretical test presented above is a suggestion to the method of determining the reservoir parameters if it is necessary in further study.

AC: In section 3.4, the following paragraph is inserted after p.10 l.8: "As the objective of our study is to explore the value of the new information brought by the high resolution watershed descriptions, the parameters were determines so that on a 1/2°grid and using the finest HTU decomposition, the routing scheme reproduces the quality of the inter-annual variability and annual cycle of the coarse resolution version. This provides a baseline against which the impact of the degradation of the HTU resolution can be evaluated on various grids."

3. The added value of the DEM and flow direction from the HydroSHEDS

RC: Many results are presented at the monthly time scale. First thing I wondered is the added value of the high resolution routing for the monthly timescale, because in principle all that routing does is delaying the water over time, but given the size of the basins, this effect is expected to be limited on the monthly time scale. So, then the added value of the improved routing is basically only in better delineation of the basins compared to the coarse resolution scheme, for which a complex DEM and flow direction scheme is not necessary. Could you respond to that?

[Figure]

**Figure 1.** Mean annual cycle of river discharge (left column) and discharge anomalies (right column) during period of 1979-2013 simulated by the new RRS with old reservoir parameters (red line) and new reservoir parameters (cyan line), in comparison with observation (black line). Three station are investigated, includes Roma - the Tiber river (top panel), Beaucaire - the Rhone river (middle panel) and Pontelagoscuro - the Po river (bottom panel). Validation metrics are shown in legend boxes includes: NS - Nash-Sutcliffe coefficient, CC - the Pearson correlation coefficient (CC), RSR - the ratio of the Root Mean Square Error to the observation standard deviation.

**AR:** *The major function of the routing scheme is to transfer the water from continent to its outlet at ocean or lake with proper time delay. In the routing scheme presented in this study, the time delay is controlled by the slope index which is calculated based on the complex DEM. The construction of river network for varying spatial resolution of the ORCHIDEE model (e.g. from 1º to 1/8º) is assisted by the flow direction. Therefore, the elevation and flow direction from HydroSHEDS is necessary for this routing scheme. The analysis of results is firstly presented at the monthly time scale. Certainly, the construction of river network based on a 1 km hydrography data allows to simulate the streamflow at higher time scale such as weekly or daily. This due to the fact that the capability of representation a river network with a lot of HTUs in the routing scheme as well as the time delay in transferring through these HTUs can reproduce the streamflow variation at frequency of daily or weekly.*

**AC:** Considering all comments from both reviewers, the section of Discussion is re-written. In doing so, the role of the complex DEM and flow direction from the HydroSHEDS is emphasized in the river basin construction which allows to flexibly delay the water with various basin size. It can be found after in p.19 l.10.

4. The analysis on the daily time scale

**RC:** The analysis on the daily time scale (e.g. Sect 5.3) to discuss model performance seems therefore more relevant, but only shows results in terms of a flow duration curve, and from a flow duration curve it is hard to retrieve timing-information. Is it not possible to include hydrograph-information to demonstrate the simulations versus observations directly?

**AR:** The timing-information for the analysis on the daily time scale can be retrieved from the evaluation metrics in the Table 3 which includes Pearson correlation coefficient, Nash–Sutcliffe efficiency, Cross-correlation lag time. The statis-

[Figure]

(a)                                                                (b)

**Figure 2.** Sensitivity of simulated discharge to the varying reservoir parameters based on a theoretical test at the station Beaucaire, the Rhone river (a) and station Pontelagoscuro, the Po river (b). The color shows the Pearson correlation coefficent of simulated discharge between old and new routing scheme.

[Figure]

(a)                                                                (b)

**Figure 3.** Taylor diagram investigates the sensitivity of simulated discharge to the varying reservoir parameters based on a theoretical test at the station Beaucaire, the Rhone river (a) and station Pontelagoscuro, the Po river (b). The color shows the Pearson correlation coefficent of simulated discharge between old and new routing scheme in corresponding to the experiments in Figure 2. The black star in the Taylor diagram denotes the result from the old routing scheme. The gray arcs are the axis for root mean square error.

tical metrics to measure differences between time-series is a more clever way for a long simulation period (1979-2013) than plotting daily series. Moreover, there are still a lot of required improvement for the routing scheme (e.g. representation of flood plain, irrigation, dams) if we want to simulate accurately the magnitude and variation of daily streamflow. The daily time series of river discharge are more affected by water management.

5      **AC:** No modification in the article.

5. The reconstruction of the conclusion

   **RC:** The current conclusion (sect 7) consists of one paragraph with the results, while the second paragraph is basically only recommendations. I think it would improve the strength of the paper to not end with mentioning everything that is missing still, but describe what has been added with this rrs. Furthermore, the results of section 4 and 5 could be touched upon in the conclusion.

   **AR:** Thank you for the comments from the reviewer. The first idea of the authors is to recall the improvement of the new routing scheme in the Discussion section then to give perspective of further development in the Conclusion section. The authors will be reorganize the arguments between section 4, 5 and 7.

   **AC:** The conclusion is modified as the suggestion from the reviewer. Please consider p.20, p.21 l1-7.

The corrected 06 minor points are:

1. The duplicate of figure caption

   **RC:** General; there is a tendency throughout the manuscript to describe the figure legend in the main text (see e.g. p.8 l.15). This is not necessary and makes the manuscript less nice to read. Describe the legend in the figure caption and the conclusions from the figure in the main text.

   **AR:** Thank you for your comments. The authors will remove the redundant description of the figure legend in the main text.

   **AC:** The description of the figure legend are removed from the main text (see p.11 l.1-2, l.18, p.13 l.2, l.22).

2. The confusion of neglected anthropogenic processes in the ORCHIDEE model

   **RC:** As not being very familiar with the models discussed in the Introduction, I don't see the logic between the section starting on p.2 l.7 and the section on p.2 l.20. The first section says: ".. redistribution in turn can feed other processes in the LSM (floodplain evaporation, irrigation). Then it continues to describe how post-processing routing schemes neglects feedback interactions between river discharge and soil hydrology. In the second section, the ORCHIDEE routing is described, but as far as I could find in the manuscript, this routing also does not account for floodplain evaporation, irrigation, or river discharge / soil hydrology interaction. Therefore this is confusing to the reader.

   **AR:** Thank you for the comment from the reviewer. The arguments related to the representation of irrigation and flood plain now only discussed in the Discussion part to prevent confusion from readers. The representations of floodplain and irrigation exist in the earlier version of the ORCHIDEE model. These parameterizations are switched off because their hypothesis are incompatible with the high resolution. They will be revised and added again once the high resolution routing is operational. For example, in the earlier version of the routing scheme in the ORCHIDEE model, the parameterization of irrigation and floodplain has already been integrated (De Rosnay et al., 2003; Guimberteau et al., 2012b; d'Orgeval et al., 2008; Guimberteau et al., 2012a). But these representations of human processes or flood plains were based on a hypothesis that HTU were rather large i.e. at scale of $1/2^o$. These parameterizations need to be revised in order to work with high resolution descriptions of the river basins. For more realistic modeling of irrigation operation in the ORCHIDEE model, the new digital global map of irrigation areas should be implemented such as the newest version from Siebert et al. (2015) with the horizontal resolution of 10 $km$.

   **AC:** The information of the floodplain and irrigation representation in the earlier version of the ORCHIDEE model is removed in p.3 l.13 and p.6 l.22. Then more arguments of this point are discussed in the Discussion part after p.18 l.20.

3. The color of the Figure 1.

   **RC:** Figure 1: the red colour appears in the legend, but is also used to indicate the regions of this study. It is unclear if the red colour then still refers to the legend colour too.

   **AR:** The authors will use another color palette without red color for displaying the area of river catchments. The red color is still used to emphasize the 12 researched rivers.

   **AC:** The color of the Figure 1 in the manuscript is modified as the Figure 4 below (also see p.5).

[Figure]

**Figure 4.** Extended simulation domain. The main watersheds are colorized as a function of maximum upstream area ($km^2$). They were extracted for an ORCHIDEE resolution of $1/4°$, with a threshold HTU number of 50. The twelve studied river basins are colored in red. They are numbered from 1 to 12, and the corresponding names are given in Table **??**. The river network is plotted in blue based on the dataset from the Generic Mapping Tools (http://gmt.soest.hawaii.edu). River names are Rhône, Ebro, Po, Chelif, Maritsa, Moulouya, Ceyhan, Tiber, Adige, Shkumbinit, Devollit, Var corresponded to number from 1 to 12, respectively.

4. The representation of irrigation and floodplains in the ORCHIDEE model

   **RC:** p.6 l.7: it is unclear how this scheme allows for irrigation withdrawal and flood plains. Perhaps elaborate on that. Furthermore, it is correct that irrigation is not accounted for in this study, right? If the routing allows for that, why was it excluded in a application in a region where irrigation is expect to play an important role?

   **AR:** Yes, the irrigation is not accounted for in this study. The routing allows for that but only for the earlier version which is limited by river map at the $1/2°$ resolution. For the new routing scheme, it is required a lot of modification in the routing scheme in order to integrate the irrigated area map to simulate the irrigation operation. Indeed, this study not only emphasizes the importance of the irrigation representation in the routing scheme but also provides the foundation for further integration of irrigation in the routing scheme.

   **AC:** Due to the modification in the Introduction, the aforementioned argument in p.6 l.7 is removed.

5. The spectrum analysis

   **RC:** p.14 l. 28 I am not familiar with the power spectrums discussed here. Perhaps some more information on this methodology can be provided, and what the implications of the results are.

   **AR:** The authors will provide more arguments with references in order to give more information on the spectrum analysis. Spectral analysis is a robust method to reflect the space-time multi-scale variation of both rainfall and runoff processes over wide ranges of basin size (i.e. from five to two millions square kilometers, Pandey et al. (1998)). It also possibly suggests physical explanations for time scale dependant relationship. For example, a 3 to 6 years oscillation typical of El Niño Southern Oscillation variability is observed in monthly river discharges of four Atlantic large rivers (Labat et al., 2005). Moreover, the spectral approach facilitates the diagnosis and development of hydrological models

(i.e. in the case study of the Thames basin, United Kingdom, Weedon et al. (2015)).

**AC:** More arguments about power spectrum analysis are provide in the paragraph starts after p.15 l.31:"It is a robust method to analyse the multi-scale temporal variations in the various variables contributing to the river discharge. As has been shown by Weedon et al. (2015), it facilitates the diagnosis and development of hydrological models. In the power-spectra trend lines for high frequencies (Variation faster than 30 days) and low frequencies (slower than 30 days) are also plotted".

6. A strong statement in the Discussion

   **RC:** p.17 l. 15 I don't see how the results support this (strong) statement.

   **AR:** The argument which the reviewer mentioned is: "The fact that the new RRS explicitly accounts for the higher quality topographical information from HydroSHEDS probably compensates the disadvantage of using simple reservoir parameters for all rivers, which is a legacy from the old RRS." Higher resolution topographical information was brought to the model without changing the simple reservoir based flow model, in order to be able to evaluate its impact on the quality of the model independently of the hypothesis in the flow model. In a second stage the flow model can be improved and floodplains or irrigation added. They will then take full advantage of the improved topographic information.

   **AC:** The statement is refined as: "The new RRS which explicitly accounts for the higher quality topographical information from HydroSHEDS improves the simulated discharge although we use a simple reservoir model for the flow and the same parameters for all rives, which is a legacy from the old RRS."

**References**

[revised manuscript text omitted]